



# Towards an objective assessment of climate multi-model ensembles. A case study in  the Senegalo-Mauritanian upwelling region

Juliette Mignot[1], Carlos Mejia[1], Charles Sorror[1], Adama Sylla[1,2], Michel Crépon[1] and Sylvie Thiria[1,3].

[1] IPSL-LOCEAN, SU/IRS/CNRS/MNHN, Paris, France

[2] LPAO-SF, ESP, UCAD, Dakar, Sénégal

[3] UVSQ, F-78035, Versailles, France

*Correspondence to*: Juliette Mignot (Juliette.mignot@locean-ipsl.upmc.fr)

**Abstract.** Climate simulations require very complex numerical models. Unfortunately, they typically present biases due to parameterizations, choices of numerical schemes, and the complexity of many physical processes. Beyond improving the models themselves, a way to improve the performance of the modeled climate is to consider multi-model averages. Here, we propose an objective method to select the models that yield an efficient multi-model ensemble average. We used a neural classifier (Self-Organizing Maps), associated with a multi-correspondence analysis to identify the models that best represent some target climate property. One can then determine an efficient multi-model ensemble. We illustrate the methodology with results focusing on the mean sea surface temperature seasonal cycle over the Senegalo-Mauritanian region. We compare 47 CMIP5 model configurations to available observations. The method allowed us to identify a performing multi-model ensemble by averaging 12 climate models only. Future behavior of the Senegalo-Mauritanian upwelling was then assessed using this multi-model ensemble.

## 1- Introduction

In this study we present a methodology aiming at selecting a coherent sub-ensemble of the models involved in the 5th Climate Model Intercomparison Project (CMIP5) that best represents specific observed characteristics. The analysis is performed on the capacity of the models to



represent the seasonal cycle of the sea surface temperature (SST) in the region of the Senegalo-
Mauritanian upwelling off the west coast of Africa.
The Senegalo-Mauritanian upwelling has focused increasing attention over the recent
years. It presents an important seasonal cycle associated with mesoscale patterns whose
variability has been recently studied by several oceanographic campaigns (Capet et al., 2017;
Faye et al., 2015; Ndoye et al., 2014). The very productive waters associated with the upwelling
have a strong economic impact on fisheries in Senegal and Mauritania, and a crucial societal
importance for local populations. It is therefore of importance to predict the evolution of the
dynamics and the physics of the upwelling in the forthcoming decades due to the effect of climate
warming and its consequences on biological productivity which may impact the fisheries. The
most common way to predict the evolution of the climate is to run climate models, which include
fully coupled atmosphere-ocean-cryosphere-biosphere modules. Because of their quite low
resolution and the fact that they use different parameterizations of the physics, numerical
schemes and sometimes include or neglect different processes, these models have some marked
biases in specific regions. They also have different responses to an imposed increase of
atmospheric greenhouse gases, which partly explain their mean climate biases. This variety of
models allows us to assess the uncertainty of present climate representation when compared to
observations and, by studying their dispersion, to roughly estimate the uncertainty of the response
to future climate change. For several generations of climate models, it has been shown that the
multi-model average for a variety of variables mostly agrees better with observations of present
day climate than any single model, and that the average also consistently scores higher in almost
all diagnostics (Gleckler et al., 2008; Lambert and Boer, 2001; Phillips and Gleckler, 2006;
Pincus et al., 2008; Reichler and Kim, 2008; Santer et al., 2009; Tebaldi and Knutti, 2007).
Several studies also suggest that the most reliable climate projection is given by a multi-model
averaging (Knutti et al., 2010), rather than averaging different projections performed with a
single model run with different initial conditions for example. This result relies on the assumption
that if choices of parameterizations, specific numerical schemes, are made independently for each
model, then the errors might at least partly compensate, resulting in a multi-model average that is
more skillful than its constitutive terms (Tebaldi and Knutti, 2007). The significant gain in
accuracy can be explained by the fact that the errors specific to each model compensate each
other in the averaging procedure used to build the multi-model. However, the number of GCMs





available for climate change projections is increasing rapidly. For example, the CMIP5 archive
(Taylor et al., 2012), which was used for the fifth IPCC Assessment Report (IPCC, 2013),
contains outputs from 61 different GCMs and 70 contributions are expected for CMIP6.
Nevertheless, these models constitute a fully independent ensemble (e.g. (Masson and Knutti,
2011). It thus becomes possible and probably needed to select and/or weight the models
constituting such an average. Recent work has suggested that weighting the multi-model
averaging procedure could help to reduce the spread and thus uncertainty of future projections.
Such an approach has been applied extensively to the issue of climate sensitivity (Fasullo and
Trenberth, 2012; Gordon et al., 2013; Huber and Knutti, 2012; Tan et al., 2016). Valuable
improvement of models selection has also been found in studies of the carbon cycle (Cox et al.,
2013; Wenzel et al., 2014), the hydrological cycle (Deangelis et al., 2015; O'Gorman et al.,
2012), the Antarctic atmospheric circulation (Son et al., 2010; Wenzel et al., 2016), extratropical
atmospheric rivers (Gao et al., 2016), atmospheric and ocean heat transports (Loeb et al., 2015),
the European temperature variability (Stegehuis et al., 2013) and temperature extremes (Borodina
et al., 2017).

The present paper is dedicated to the elaboration of an objective method to select models

according to their performance over the Senegalo-Mauritanian upwelling region, with the aim of
constructing an efficient climate multi-model average together with its related confidence interval
in order to anticipate the effect of climate warming by the end of the century in this region.  This
upwelling is very intense and presents a well-marked seasonal variability. Its intensity is stronger
in boreal winter and it disappears in summer with the northward progression of the ITCZ.  Due to
the enrichment of the sea surface layers with nutrients upwelled from deep layers, it drives an
important phytoplankton bloom that is observed on ocean color satellite images (Demarcq and
Faure, 2000; Farikou et al, 2015). The maximum intensity of this bloom occurs in March-April
(Farikou et al., 2015; Faye et al., 2015; Ndoye et al., 2014). This upwelling lies at is the southern
end of the Canarian upwelling system. North of 20°N, which is the northernmost latitude reached
by the ITCZ, the seasonality of the upwelling is much weaker. It is forced by the trades, which
are more intense in summer. Consequently, the Senegalo-Mauritanian upwelling is characterized
by a very specific seasonality which is observed on satellite SST (Demarcq and Faure, 2000;
Sawadogo et al., 2009). Sylla *et al.* (2019) have recently showed that the intensity of the SST





seasonal cycle along the coast of Senegal and Mauritania was a good marker of the upwelling in
climate models.
The paper is articulated as follows: section 2 presents the different climate models and
the climatological observations used in the study, together with the region of interest. The
classification method is described in section 3 for the extended region and results are discussed.
Section 4 investigates the results of the method applied over a smaller area, more focused over
the region of interest. Section 5 uses the two multi-model clusters defined in sections 3 and 4
respectively to describe the representation of the Senegalo-Mauritanian upwelling and its change
under global warming. Conclusions are given in section 6.

### 98   2- Models and region of interest

### 99 2.1 Data

This study is based on the CMIP5 (Coupled Model Inter-comparison Project Phase 5) database.
We used the output of the 47 simulations listed in Table 1. The models have been evaluated over
the historical period defined as [1975-2005] by comparing their output to observations. The mean
seasonal cycle of SST over this period is constructed for each model grid point. When several
members of historical simulations are available for a same model configuration, they are
averaged together. However, this has practically no impact on the estimated mean seasonal cycle
(not shown). The mean climatological cycle of the CMIP5 models under study is evaluated
against the ERSST_v3b data set (Smith et al., 2008), averaged over the same time period. This
data set is used as the target to be reproduced and is denoted "Observation field" hereinafter. In
order to deal with data at the same resolution, all model outputs as well a observation fields were
been regridded on a 1-degree resolution regular grid prior to analysis.
In section 5, the models' selections are used to characterize the response of the upwelling to
climate change. This response is characterized in terms of SST but also wind intensity. For this,
the simulated wind stress is compared to the QUICKSCAT product (https://podaac.jpl.nasa.gov)
and the models are evaluated over the period [1985-2005].

### 116 2.2 The Senegalo-Mauritanian upwelling region





In the present research, we evaluated the ability of the different climate models to represent the
Senegalo-Mauritanian upwelling. Following Sylla et al. (in rev.), we consider the intensity of the
seasonal cycle of the SST as a marker of the upwelling variability and localization. This variable
is shown in Fig. 1 for the eastern tropical Atlantic. This figure confirms that the Senegalo-
Mauritanian coast stands out with a very strong seasonal SST cycle as compared to what is found
at similar latitudes in the open ocean. This results from the cold SST generated by the strong
winds occurring in winter. The Senegalo-Mauritanian upwelling is confined in a small region of
the order of 100km off the western coast of Africa. It is part of a complex and fine scale regional
circulation system recently revisited by Kounta *et al.*, (2018). Since the grid mesh of most of the
climate models is of the order of 1° (~100km), this regional circulation is thus poorly resolved,
and this pleads for a relatively large-scale analysis of the upwelling representation in climate
models. The Senegalo-Mauritanian upwelling is also embedded in a large scale oceanic
circulation pattern, encompassing the North Equatorial Counter Current flowing eastward in the
southern part of the region and the return branch of the subtropical gyre in the northwestern part.
Therefore, we will firstly study the representation of the SST seasonal cycle intensity in the
different climate models over a relatively large region that includes part of the Canary current in
the North and the Guinea dome in the South. The so-called "extended region" is defined by a
rectangular box extending from 9°W to 45°W and from 5°N to 30°N (Fig. 1). In a second step,
we will proceed to the same analysis and classification of the models within a much more
focused (hereafter zoomed) region, namely [16°W-28°W and 10°N-23°N] (Fig. 1). All the results
below will be first shown for the extended region. Comparison with the focused region will be
done in section 4.
**3 - Classification of the climate models over the extended upwelling region**
**3.1 The methodological approach**
The first step of the methodology is to decompose the selected region in different classes by
using a neural network classifier, the so-called Self Organizing Map (SOM; Kohonen, 2013).
This algorithm constitutes a powerful nonlinear unsupervised classification method. It has been
commonly used to solve environmental problems (Hewitson and Crane, 2002; Jouini et al., 2013,
2016; Liu et al., 2006; Reusch et al., 2007; Richardson et al., 2003). The SOM aims at clustering
vectors of a multidimensional database (***D***) into classes represented by a fixed network of



neurons (the SOM map). The self-organizing map (SOM-map) is defined as an undirected graph,
usually a 2D rectangular grid. This graphical structure is used to define a discrete distance
(denoted by $\delta$) between the neurons of the map and thereby identify the shortest path between
two neurons. Moreover, SOM enables the partition of **D** in which each cluster is associated with a
neuron of the map and is represented by a prototype that is a synthetic multidimensional vector
(the referent vector **w**). Each vector **z** of **D** is assigned to the neuron whose referent **w** is the
closest, in the sense of the Euclidean Norm (EN), and is called the projection of the vector **z** on
the map. A fundamental property of a SOM is the topological ordering provided at the end of the
clustering phase: two neurons that are close on the map represent data that are close in the data
space. In other words, the neurons are gathered in such a way that if two vectors of **D** are
projected on two "relatively" close neurons (with respect to $\delta$) on the map, they are similar and
share the same properties. The estimation of the referent vectors **w** of a SOM and the topological
order is achieved through a minimization process using a learning data set base, here from the
observations. The cost function to be minimized is of the form:

$$J_{SOM}^{T}(\chi, W) = \sum_{z_i \in D} \sum_{c \in SOM} K^T(\delta(c, \chi(z_i))) \|z_i - w_c\|^2$$

where $c \in SOM$ indices the neurons of the SOM map, $\chi$ is the allocation function that assigns
each element $z_i$ of **D** to its referent vector $w_{\chi(z_i)}$ and $\delta(c, \chi(z_i))$ is the discrete distance on the
SOM-map between a neuron c and the neuron allocated to observation $z_i$. $\boldsymbol{K^T}$ a kernel function
parameterized by $T$ (where $T$ stands for "temperature" in the scientific literature dedicated to
SOM) that weights the discrete distance on the map and decreases during the minimization
process. At the end of the learning process, the classification can be visualized onto the SOM-
map and interpreted in term of geophysics.
**3.2 - Classification of the observations**
In the present problem we chose to classify the annual cycles of the SST seasonal anomalies
observed in the Senegalo-Mauritanian upwelling. The study was made over the "extended
region" constituted of $25 \times 36 = 900$ pixels , but this enlarged region covers a part of the African
continent and 157 pixels are in fact over land. That means that we have truly 743 ocean pixels to deal
with. We consider the time-period of 30 years [1975 to 2005] extracted from the ERSST-V3b
database. For a given grid point and a given year and month, the monthly anomaly is the SST of





the pixel for which we have subtracted the mean of the considered year. The climatological mean
of the anomaly is then computed for each grid point by averaging each climatological month over
the 30 years. Thus, the learning data set $D$ is a set of 743 twelve-component vectors $z$, each
component being the mean monthly anomaly computed as above. We denote "SST Seasonal
Cycle" the vector $z$ in the following.
We used a SOM-map to summarize the different SST seasonal cycles present in the "extended
region". We found that 120 prototypes (or neurons) can accurately represent the 743 vectors of $D$.
This reduction (or vector quantization) is made by using a rectangular SOM-map of $30 \times 4$
neurons.
We then reduced the number of neurons in order to facilitate their interpretation in terms of
geophysical processes. For this, we applied a Hierarchical Ascendant Clustering algorithm
(HAC) using the Ward dissimilarity (Jain and Dubes, 1998). We grouped the 120 neurons into a
hierarchy that can contain between 120 and 1 clusters. Then the different classifications proposed
by the HAC were applied to the geographical region: each "SST Seasonal Cycle" of each grid
point of the region is assigned to a neuron and consequently to a cluster (assignment process),
thereby defining the so-called region-clusters. The problem is then to choose a number of clusters
that adequately synthesizes the geophysical phenomena over the region. This was done by
looking at the different possible classifications and choosing one representing the major
characteristics of the upwelling region. In Fig. 2a, we observe that when we partition the SOM in
7 clusters, the associated 7 region-clusters are constituted of contiguous pixels in the geographic
map, and that two clusters (6, 7) are within the upwelling region and present a well-marked
seasonal cycle. For each region-cluster, we can estimate the associated standard deviations (STD
hereinafter) by processing all the data assigned to its associated neurons with a standard statistic
algorithm. The typical SST climatological cycles for each region-cluster are presented in Fig. 2b
together with their related error bars. We note that the region-clusters are well identified, their
typical climatological annual cycles of SST anomalies being well separated. Furthermore, the 7
region-clusters are spatially coherent and have a definite geophysical significance.
For the extended region under study, 7 therefore appears to be an adequate cluster number, since
this number allows a clear partition of the clusters on the HAC decision tree on the one hand, and
permits to assign a clear physical significance to each region-cluster on the other hand. Let us





now describe these clusters according to their physical significance with respect to the
geographical region: the Senegalo-Mauritanian coastal upwelling is associated with clusters 7 and
6. Cluster 2 corresponds to deep tropical waters associated with the equatorial Countercurrent.
Cluster 1 corresponds to surface waters of the Gulf of Guinea. Cluster 3 corresponds to the
offshore tropical Atlantic, and cluster 5 has extratropical characteristics. Cluster 4 is transition
between 3 and 5. As expected, the equatorial regions (clusters 1 and 2) have a very weak seasonal
cycle, which increases towards the extratropics (clusters 3 to 5). The upwelling regions (clusters
6 and 7) are characterized by an exceptionally strong seasonal variability.
**3.3 – Analysis of the different climate models**

The aim is now to find the model(s) that best fit the "Observation field". A heuristic
manner is to compare the pattern of the different region-clusters of the CMIP5 models with
respect to those of the "Observation field" through a sight evaluating process. This kind of
approach has been proposed in (Sylla et al., 2019), and one indeed immediately sees that some
models better fit the "Observation field" than others. But this method remains very subjective.

In the following, we present a more objective approach. We use the previous
classification to objectively estimate how each CMIP5 model represents the "Observation field"
and its seven region-clusters. For this, we projected the SST annual cycle of each CMIP5 model
grid point of the extended region onto the SOM learned with the observations (section 3.2) using
the assignment procedure described in this section. Each grid point thus corresponds to a cluster
of the SOM and is represented on the geographical map by its corresponding color.  Doing so, we
can represent each CMIP5 model by the geographical pattern of the 7 clusters partitioning the
SST seasonal cycle of its grid points. The geographical maps representing the 47 models and
their associated clusters are plotted in Fig. 3. This graphical visualization is easier to compare
than the original characteristics (amplitude and phase) of the annual cycle at each grid point of a
model since each grid point can only take one discrete value among seven. This representation
immediately allows identifying the model biases and the models that best reproduce the cluster-
regions identified in the observations.

For a more quantitative assessment, we counted the number of grid points of a region-
cluster for a given CMIP5 model matching the same region-cluster of the "Observation field".
We then computed the ratio between that matching number and the number of pixels of the





region-cluster of the considered model. That number is noted in the color-bar for each region-
cluster in Fig. 3. We denote Rmi the ratio for the region-cluster i and the model m, where
i = 1, …, 7 is the number of the region-cluster and m = 1, …, 47 is the number of the model (see
table 1). We note that Rmi ≤ 1.  Doing so, each model m is represented by a 7-dimensional vector
Rm, each component being the ratio of a region-cluster.  We estimated the total skill of a model
by averaging the 7 ratios. Note that this procedure gives the same weight to each region-cluster
whatever its number of grid point and its proximity with the upwelling region. In the following
the skill is presented as a percentage, the higher the skill, the better the fit. In Fig. 3, the 47
CMIP5 models are ranked by their total skill, which is indicated above each panel beside the
model name. The model skills are very diverse, ranging from 79% to 28%. This Fig. also shows
that the models presenting the best total skill are also those representing thoroughly the upwelling
region. Some models represent the large-scale structure in the eastern tropical Atlantic (region-
clusters 3, 4, 5) very well but not the upwelling (33-GISS-E2-R and 34-GISS-E2-R-CC for
example). Others represent pretty well the upwelling region-clusters (region-clusters 6 and 7), but
not the large-scale structures of the SST seasonality (13-CSIRO-Mk-3-6-0, 6-CMCC-CESM for
example). None of this type of models is ranked among the best models, let us say with a score of
more than 60%. As indicated above, this representation gives a very synthetic view of the
structure of the seasonality of the SST in each of the models, potentially a very useful guide for
climate modelers to identify rapidly major biases.

**3.4 – Categorial analysis of the CMIP5 climate models**

In order to further progress in the selection of the models, the 47 climate models and the

Observation field were then analyzed by using a Multiple Correspondence Analysis (MCA in the
following). MCA is a multivariate statistical technique that is conceptually similar to principal
component analysis (PCA in the following), but applies to categorical rather than continuous
data. Similarly as PCA, it provides a way of displaying a set of data in a two-dimensional
graphical form. In the following, we applied a MCA analysis to the (47 × 7) matrix R = [Rmi]
whose elements represent the skills of the clusters of the models shown in front of the color bars
in Fig. 3:  the rows represent the different models, the columns the seven region-clusters. We
found that the first two principal axes of the MCA provide 71% of inertia. In Fig. 4, we show a





projection of the models on the plane defined by the first two principal axes (each model being
represented by a small circle). Moreover, we projected the observation field (green diamond) on
that plane as a supplementary individual. The shorter the distance between two models, the more
similar the distribution of their region-cluster skills. On that plan, the seven clusters of the
observation field are represented by purple squares. Proximity between a model and a region-
cluster leads us to affirm that this region-cluster is well represented by that model. Clearly, some
models adequately represent the southern part of the extended region (region-clusters 1, 2 or 3),
where the SST seasonal cycle is weak, and are very distant from the upwelling regions (region-
cluster 6 and region-cluster 7) whose large SST cycle is poorly reproduced. In this group of
models, one recognizes the model 16-IPSL-CM5A-MR, at the extreme bottom of Fig. 4, close to
region-clusters 4 and 5, consistently with Fig. 3. At the other end of this group of models, the
model 23-HadCM3 for example is located very close to the region-cluster 1. Fig. 3 indeed shows
that most of its grid points over the region of interest have a seasonal cycle resembling the one
found in the offshore tropical ocean. Another group of models is located in the center of this plan,
thus at an optimal distance of each of the observed regions-clusters, and not far from the overall
position of the observations (diamond). We recognize in this group of models, those that have a
high skill in Fig. 3. The positioning of the observations (diamond in Fig. 4) with respect to the
models indeed allows selecting those that best represent the Observations field. The
representation given in Fig. 4 allows understanding the drawback of the different models with
respect to the 7 Modes of SST-cycles.

As indicated in the introduction, the main objective of the methodology is to select an
ensemble of models that represents at best the upwelling behavior with respect to the
observations and to use this ensemble to predict the impact of climate change in the Senegalo-
Mauritanian upwelling with some confidence. The problem is now to determine a subset of
models that can adequately represent the observations, as the number of models is small enough
we choose to cluster them by HAC according to their projections onto the seven axes provided by
the MCA, and select the optimal jump in the hierarchical tree (Jain and Dubes, 1998).

Doing so, we obtain four homogeneous groups which are well separated (group 1, 2, 3, 4).
They are plotted with different colors in Fig. 4. Clearly, the models are clustered with respect to
the region-clusters they best represent. We denote Model-group 1, Model-group 2, Model-group





3, Model-group 4 these multi-model ensembles hereinafter. Model-group 4 represents the
observations and the upwelling region-clusters at best.

For each group, we computed a multi-model average whose outputs are the mean of the

outputs of its different members and we analyzed it according to the same procedure (projection
of the SST-seasonal Cycle and assignment to a region-cluster) used for each individual model.
Besides we introduced the full multi-model average (Model-All in the following), which is the
multi-model ensemble which averages the 47 CMIP5 model outputs. Model-All was also projected
in the MCA plane and it is represented by a red star in Fig. 4. Comparison of the four model-groups
with Model-All and the observations are presented in Fig. 5. This figure visually highlights the
dominance of Model-group 4 for the reconstruction of the SST seasonal cycles of the different
region-clusters for the extended region. This is particularly clear for region-clusters 6 and 7,
which are those located in the upwelling region (Fig. 2). Model-group 3 seems to group models
characterized by an equatorward shift of the main structures, since the region-cluster 1 of tropical
waters is not reproduced and Region-clusters 4 and 5 of extratropical waters are overestimated.
Fig. 4 indeed shows that this Model-group is very close to the Regions-clusters 4 and 5, which
correspond to the extratropical and the transition geographical regions. Model-group 2
misrepresents the region of the Canary upwelling. Model-group 1 overestimates the SST seasonal
cycle in all the tropical open Atlantic. These two last model-groups overestimate the region-
Cluster 1, again consistently with their position in Fig. 4. A detailed physical interpretation of the
Model-groups is nevertheless beyond the scope of this paper. Clearly Model-All represents the
SST seasonal cycle of the off-shore ocean, but it proposes a very poor representation of the
upwelling region.

Two models (models 7 and 25) have a better skill than Model-group 4 and Model-All.

These two models are very close to the observations on the first two axes of the MCA (Fig 4). It
is easily seen that Model-group 4 and the projection of Model-All on this plane is farther than
that of model 7 and model 25 from the observation projection. This explains the lower
performance of these two multi-models as compared to models 7 and 25. In the present case, the
method permits to determine the best models (model 7 and model 25) and to outline the best
multi-model (Model-group 4) whose skill is better than any model with a probability of 95%
(number of models whose skill is smaller than the skill of Model-group 4 with respect to the total





number of models). Projection of the models on the other planes of the MCA analysis should
confirm this interpretation. One could then question the use of Model-group 4 rather than model
7 or model 25 individually. Furthermore, we argue that multi-model averages are in general more
robust for all sorts of climate studies than the use of a single model that can have good
performance for a very specific set of constraints but not for neighboring ones. The following
section will partly justify this point.
**4 - Classification of the climate models over a zoomed upwelling region**
The classification presented above relies largely on the ability of the models to represent
the off-shore seasonal cycle of the SST. In the following, we propose to test the classification
over a much more reduced area in order to focus the analysis on the upwelling area. This
"zoomed upwelling region" is shown in Fig. 1. As for the extended region, we partitioned the
observations of the zoomed upwelling region with a SOM (ZSOM in the following) followed by
a HAC. We obtained new region-clusters denoted ZRegion-clusters. Fig. 6 shows the four
ZRegion-clusters obtained from ERSSTv3b observations together with their associated mean
SST-Seasonal Cycle. Again, the ZRegion-clusters are spatially coherent. The upwelling area is
now decomposed into three ZRegion-clusters (ZRegion-clusters 2, 3, 4). This new decomposition
thus refines the study performed for the extended region: ZRegion-cluster 1 represents the
offshore ocean: its grid points typically have a SST seasonal cycle amplitude of 4°C, very similar
to Region-cluster 4 in the classification performed over the extended region (Fig. 2). ZRegion-
cluster-4 nicely identifies the core of the Senegalo-Mauritanian region, with grid points
characterized by the greatest amplitude of the SST seasonal cycle of the domain: typically 6.5°C.
It is interesting to note that an additional upwelling ZRegion-cluster (ZRegion-cluster 3) appears
south of ZRegion–cluster 4. Indeed, several studies have shown that the Cape Verde peninsula,
located around 15°N, separates the upwelling region into two distinct areas having a different
behavior north and south of this peninsula (Sirven et al. sub.,(Sylla et al., 2019)). The location of
the separation between ZRegion-cluster 3 and 4 is determined with some uncertainty due to the
coarse resolution (1°) of the ocean models. ZRegion-cluster 3 is marked by a time shift of the
seasonal cycle: the warmest season seems to occur somewhat one month earlier than in the other
regions as clearly seen in Fig. 6 (left panel, yellow curve in June). Due a classification done in a much
larger region, such characteristic does not appear in the study over the extended area study. The



physical interpretation of the SST seasonal cycle of this ZRegion-cluster is beyond the scope of
the present study, but one can suspect a role of the ITCZ seasonal migration, covering these grid
points earlier than further north. Finally, ZRegion-cluster 2 is a transition between the large scale
ocean and the upwelling region. As for the extended region, we applied a MCA analysis to the
$(47 \times 4)$ matrix $R = [Rmi]$ whose elements represent the skills of the four clusters (i) of the 47
models. This MCA was followed by a HAC leading the definition of five ZModel-groups. The
members of each group are given in appendix. Fig. 7 shows the ZRegion-cluster obtained in the
zoomed area by projecting these five ZModel-groups and Model-All model on the ZSOM and
their associated performances. ZModel-group 1 is the least performing one: only 25% of the grid
cells fall in the same class as for the observations. The structure of this model-group shows that it
is characterized by an homogeneous amplitude of the seasonal cycle over the whole domain,
suggesting a largely reduced upwelling: only one grid point at the coast has an enhanced SST
seasonal cycle as compared to the large scale tropical ocean. ZModel-group 2 is the best
performing one: 66% of the grid points are assigned to the correct class and the general picture
indeed represents a four-class picture fairly consistent with the observed structure (Fig. 6).
Important biases yet remain. In particular, the ZRegion-clusters 2 and 4 characterizing the
upwelling extend too far offshore. The three other ZModel-groups are intermediate. A relatively
reduced upwelling area, with an underestimated SST seasonal cycle, characterizes ZModel-
groups 3 and 4. ZModel-group 5 corresponds to a shift of the upwelling region towards the north.
Model-All also shows a strongly reduced seasonal cycle, with a large amount of pixel in the
intermediate ZRegion-cluster 3 and very few in the ZRegion-cluster 4. The ZRegion-cluster 3
representing the southern part of the Senegalo-Mauritanian upwelling does not appear in the
pattern of Model-All.
We remark that all the models forming ZModel-group 2 are included in Model-group 4.
For a more precise assessment, we can also project the entire Model-group 4, identified as the
best multi-model ensemble over the extended region, on the ZSOM (Fig. 8, right). We notice that
the performance of Model-group 4 remains very high on this projection, indicating some
robustness of this multi-model ensemble. Moreover, this ensemble now outperforms the single
best model identified over the extended region (Fig. 8, right). This result gives further confidence
in the use of multi-model averages, illustrating that one single model can be very skillful over a





specific region, or for a specific analysis, but multi-model averages are more robust across
various analysis and/or regions.
**5 – Impact of climate change on the Senegalo-Mauritanian upwelling**
**5.1 Representation of the upwelling in the CMIP5 climate models clusters**
In this section, we compare the representation of the Senegalo-Mauritanian upwelling system
given by the two best Model-groups identified above (Model-group 4 and ZModel-group 2). For
this evaluation, we use two of the five indices used by (Sylla et al., 2019) to evaluate the full
database, namely the intensity of the SST seasonal cycle and the offshore Ekman transport at the
coast. The former is specific to the seasonal variability of the Senegalo-Mauritanian upwelling
system, and it has been used for the classification. The latter is more general and although it has
recently been shown to partly represent the volume of the upwelled waters (Jacox et al., 2018), it
is extensively used in the scientific literature to characterize upwelling regions (Cropper et al.,
2014; Rykaczewski et al., 2015; Wang et al., 2015). Note also that following (Sylla et al., 2019),
evaluation is performed on the period [1985-2005]. This period slightly differs from the
classification period but the SST seasonal cycle is not significantly different (not shown).

Fig. 9 compares the amplitude of the SST seasonal cycle as represented in the

observations, Model-All, Model-group 4 and ZModel-group 2 identified above. Consistently with
Fig. 5 and 7, Model-All dramatically underestimates the upwelling signature in terms of SST
seasonal cycle as compared to the observations. Model-group 4 and ZModel-group 2 yield
improved results: the area of enhanced SST seasonal cycle is larger both in latitude and
longitude, with stronger SST amplitude values. This confirms the efficiency of the selection
operated above. Nevertheless, ZModel-group 2 yields a realistic SST amplitude pattern along the
coast but it extends too far offshore. Furthermore, in ZModel-group 2, the subtropical area (in
green in Fig 9) extends too far towards the south, in particular in the western part of the basin.
The tropical area, characterized by limited amplitude of the seasonal (deep blue in Fig. 9), is
shifted to the south as compared to the observations. In other words, the large scale thermal, and
thus probably dynamical structure of the region is poorly represented in ZModel-group 2. Finally,
Model-group 4 is the least biased one.





The intensity of the wind stress parallel to the coast, inducing offshore Ekman transport
and consequently an Ekman pumping at the coast, is generally considered as the main driver of
the upwelling. We therefore also tested the representation of this driver in the different Model-
groups. The idea is to evaluate the impact of the model selection performed above on the
representation of an independent variable by the Model-groups. Fig. 10 shows the latitude-time
evolution of the meridional oceanic wind stress, considering that the coast in the studied region is
oriented approximately meridionally, so that the offshore Ekman transport is mainly zonal. Note
that in Fig. 10, southward winds have positive values so that they correspond to a westward
Ekman transport, favorable to upwelling. Panel (a) shows that the observed meridional wind
stress is, all year long, favorable to the upwelling north of 20°N. At these latitudes, it is stronger
in summer. Between 12°N and 20°N, in the latitude band of the Senegalo-Mauritanian upwelling,
on the contrary, the wind blows southward with a very weak intensity in summer and it even
changes direction in the southern part of this latitude band. It is favorable to the upwelling in
winter-spring, which explains why the Senegalo-Mauritanian upwelling occurs during this season
with a maximum of intensity in March-April  (Capet et al., 2017; Farikou et al., 2015).  The main
bias of Model-All (Fig. 10b) is that the wind stress never reverses between 12°N and 20°N. It
weakens in the southern part of the Senegalo-Mauritanian latitude band, i.e. south of the Cape
Verde peninsula (15°N), but does not become negative. North of the Cape Verde peninsula, it
blows from the north also in summer, so that the Senegalo-Mauritanian upwelling lacks of
seasonality. This bias is corrected in Model-group 4 and ZModel-group 2 (Fig. 10 , panels c and
d) that are, in this aspect, more realistic than Model-All. Model-group 4 shows a slight extension
of the time and latitude range where the oceanic wind stress reverses sign. This constitutes an
improvement. The southward wind is nevertheless too strong in winter over the [12°N-20°N]
latitude band as well as further south from December to March. These two remaining biases are
further reduced in ZModel-group 2. This latter model yields the most realistic seasonal cycle of
meridional oceanic wind stress over the latitude band under study. This is consistent with a very
localized model selection, as the wind index is itself localized along the coast.
To conclude, Model-group 4 and ZModel-group 2 perform in general better than Model-All in
reproducing the major characteristic features of the Senegalo-Mauritanian upwelling. This result
confirms the relevance of the multi-model selection we have presented above. Applying the
methodology over a relatively large region allows to better constrain the spatial extent and pattern





of the SST signature of the upwelling than the reduced area. The latter however yields a better
representation of the wind seasonality along the coast.

**5.2 Response of the Senegalo-Mauritanian upwelling to global warming.**

In this section, we examine the response of the upwelling system given by the different

multi-model groups we selected, to global warming. For this, we compared the two indices
analyzed above in present-day and future conditions. The present-day conditions are taken as
above as the climatological average of historical simulations over the period [1985-2005]. The
future period is taken as the climatological average of the RCP8.5 scenario over the period [2080-
2100]. Fig. 11 shows the difference of the SST seasonal cycle amplitude between these two
periods. The general behavior is that the SST cycle amplitude will reduce in the upwelling region.
(Sylla et al., 2019) showed that this is primarily due to a warming of the winter temperature, thus
suggesting that the upwelling signature in surface will reduce. On the other hand, this figure
shows that the upwelling signature will increase along the Canary current, which flows along the
coast of Morocco, as well as in the subtropical part of our domain. This behavior is observed in
the three multi-model ensembles. Yet, the two selected Model-groups suggest a weaker decrease
of the SST seasonal cycle in the upwelling region than the one given by Model-All. ZModel-
group 2 shows an even weaker decrease mainly confined in the southern part of the upwelling
region. This result echoes findings of (Sylla et al., 2019) based on another indicator of the
upwelling imprint on the SST: they showed that the difference between the SST at the coast and
offshore is expected to decrease more in the southern part of the Senegalo-Mauritanian upwelling
system (SMUS) than in the north . We can hypothesize that the study conducted on the reduced
area permits to separate the Senegalo-Mauritanian upwelling system into two clusters, a northern
one (ZRegion 4) and a southern one (ZRegion-3) (Fig. 7) which enables to distinguish this
specific response.

The meridional wind stress also generally weakens under climate change in the [12°N-

20°N] latitude band (Fig. 12), suggesting a general reduction of the upwelling intensity. From
December to March, this is particularly true in the southernmost region of the Senegalo-
Mauritanian band, consistently with the results of (Sylla et al., 2019). The wind pattern inferred
from the two Model-groups (Fig. 12, middle and right panels) present a higher seasonal
variability than this of Model-All (left panel). The winter reduction of the southward wind stress





is slightly more confined to the southern region in ZModel-group 2, especially at the end of the upwelling season (March-April) when the upwelling intensity is the strongest. This may be consistent with the reduced seasonal cycle in the southernmost part of the upwelling identified above.

**6 - Discussion and Conclusion**

This paper proposed an objective methodology for selecting climate models over a specific area with respect to observations and according to well-defined statistical criteria. In the present study, we have specifically checked the ability of the climate models to reproduce the ocean SST annual cycle observed in specific regions of the studied area during the period 1975-2005 as reported in the ERSST_v3b data set. These regions were defined by a neural classifier (SOM) as clusters having similar seasonal SST cycles with respect to some statistical characteristics. They correspond to ocean area having well marked oceanographic specificities.

We then checked the ability of the different climate models to reproduce the Region-clusters defined on the observation dataset with a SOM. The better a climate model fits the clusters computed with the SST observation, the better the skill of the model. We thus defined geographical regions in the different CMIP5 climate models by projecting the SST annual cycle of each model grid point onto the SOM. Each grid point is associated with a cluster on the SOM map and consequently to a Region-cluster on the geographical map. We built an objective similarity criterion by counting the number of grid points in a Region-cluster of a given model matching the same Region cluster defined by processing the "Observation field". We then computed the ratio between that matching number and the number of pixels of the Region-cluster of the model under study. We estimated the total skill of a model by averaging the 7 ratios associated with the 7 Region clusters. Note that this procedure presents the advantage to give the same weight to each region-cluster whatever its number of grid point and its proximity with the upwelling region. This procedure respects the clustering done by the SOM since the different clusters have an equal weight in the skill computation. In its present definition, the total skill is a number between 0 and 1, the higher the skill, the better the fit. Other measures of the total skill of a Model-group could nevertheless be defined depending on the objective of the study.





Such a multi-model ensemble selection indeed allows subsampling a set of models in order to
obtain a more realistic climatology over the region of interest. The response of the upwelling to
climate change given by the different multi-model ensembles is quite robust in the sense that they
give similar qualitative answers. However, a too selective ensemble of models may lead to noisy
patterns. A compromise thus has to be found between the advantage of using a large number of
models, in order to smooth biases and unrealistic patterns, or selecting the most realistic models,
with the advantage of using a small number of models in the averaging procedure, but with the
possible inconvenience of getting spurious biases.
Different criteria have been used for selecting the best models included in the multi-model
ensemble used for climatic studies. The most common parameter is the average annual variability
in the surface mean temperature of the grid points of the region under study. Besides, (Knutti et
al., 2006) used the seasonal cycle in surface temperature represented by seasonal amplitude in
temperature calculated as summer June–August (JJA) minus winter December–February (DJF)
temperature. This criterion is more informative than the annual variability in the mean
temperature since the amplitude of the seasonal variability is an important criterion characterizing
the validity of a climate model. In the present work we used a much more informative criterion
which is formed of the monthly temperature cycle represented by a 12 component vector, each
component representing the average monthly temperature of the year we consider. This new
criterion allows taking account the amplitude and the phase of seasonal variability while the
(Knutti et al., 2006) criterion takes only into account the amplitude of the seasonal variability.
More generally, (Sylla et al., 2019) extensively discussed the possible differences among the
different indices aiming at characterizing the upwelling and the need to use several of them to
have a complete understanding of this costal phenomenon. This conclusion is probably general to
any physical process of the climate system. In the present study, the model selection is only based
on one signature of the SMUS. Ongoing studies in our group investigate the possibility of
merging several indices such as SST, wind intensity and direction, ocean currents,... This
approach could also allow a selection of models based on the representation of several distinct
regional processes.
Different applications of the multi-model selection strategy proposed in the present study can be
envisaged. Firstly, from a purely modeling point of view, the projection of the models on the



SOM (or ZSOM) and the results of the HAC yield a very enlightening description of a given
model behavior in terms of region-clusters of the area under study. In our view, such a procedure
could advantageously be used by individual modeling groups to identify, analyze and therefore
hopefully reduce their model biases in a targeted region. Secondly, from a physical point of view,
an identified Model-group can be used to analyze the targeted region (here the SMUS) in term of
processes with the advantages of the multi-model mean in which the constituting models have
been selected from objective criteria. Such an application has been briefly illustrated by showing
how the selected Model-group represents an important additional characteristic of the SMUS, not
used for the selection, namely the Ekman pumping. Promising reduction of biases of the full
multi-model mean ensemble has been identified, opening perspectives for process studies based
on this sub-ensemble of the CMIP5 database. A third application of the selection lies in the
prediction of the future climate. Here, we have shown that selected multi-model ensembles may
provide a more precise description of the future behavior of the SMUS. It may nevertheless be
important to note that these conclusions are based on the assumption that the CMIP5 models
which have been selected according to their present-day characteristics, are the most reliable in
terms of future projections, which can be questioned and refined (Lutz et al., 2016; Reifen and
Toumi, 2009).
As discussed in the introduction, "model democracy", suggesting that all models should be
equally considered in multi-model ensemble is now strongly questioned (Knutti et al., 2017). The
present study proposes a promising way to improve the quality of multi-model ensemble fin
terms of model selection. Deep advances in the field of multi-model analysis and selection can be
expected from the emerging topic of climate informatics (Monteleoni et al., 2013) as it has been
shown through the present study. Artificial intelligence and machine learning may indeed provide
efficient tools to progress in making the best out of the extraordinary but imperfect tools that are
the climate models and the multi-model intercomparison efforts.

### Acknowledgments

NOAA_ERSST_V3b data provided by the NOAA/OAR/ESRL PSD, Boulder, Colorado, USA,
from their Web site at https://www.esrl.noaa.gov/psd/ The research leading to these results has
received funding from the NERC/DFID Future Climate for Africa program under the SCUS-2050
project, emanating from **AMMA‑2050** project, grant number NE/M019969/1. The authors also



acknowledge support from the Laboratoire Mixte International ECLAIRS2, supported by the
french Institut de Recherche pour le Développement. To analyze the CMIP5 data, this study
benefited from the IPSL Prodiguer-Ciclad facility which is supported by CNRS, UPMC, Labex
L-IPSL which is funded by the ANR (Grant #ANR-10-LABX-0018) and by the European FP7
IS-ENES2 project (Grant #312979)
**Code and Data availability**: The model output used for this study is feely available on the ESGF
database for example following this url: https://esgf-node.ipsl.upmc.fr/search/cmip5-ipsl/. The
SST data were downloaded from
https://www.esrl.noaa.gov/psd/data/gridded/data.noaa.ersst.v3.html and the winds data
here: https://podaac.jpl.nasa.gov . The code developed for the core computations of this study can
be found under: 10.5281/zenodo.3476724. This code allows reproducing Fig. 2, 3, 6, 7 and 8.
**Author contribution**: JM initially proposed the idea, ST and MC translated it in terms of
methodology and coordinated the method development, CS and CM developed the code and
produced the figure, CS, CM, MC, ST all contributed to the statistical analysis. AS provided the
initial definition of the upwelling index and performed the analysis under climate change that
appears in section 5. JM and MC prepared the manuscript with contributions from all the authors.





**APPENDIX**

| Model-group 1 | Model-group 2 | Model-group 3 | Model-group 4 |
|---|---|---|---|
| ACCESS1-0 | bcc-csm1-1 | FGOALS-g2 | CanCM4 |
| ACCESS1-3 | bcc-csm1-1-m | GISS-E2-H | CanESM2 |
| CESM1-CAM5 | BNU-ESM | GISS-E2-H-CC | CMCC-CESM |
| CESM1-CAM5-1-FV2 | CCSM4 | GISS-E2-R | CMCC-CM |
| CESM1-WACCM | CESM1-BGC | GISS-E2-R-CC | **CMCC-CMS** |
| HadCM3 | CESM1-FASTCHEM | inmcm4 | **CNRM-CM5** |
| MIROC-ESM | GFDL-CM2p1 | IPSL-CM5A-LR | **CNRM-CM5-2** |
| MIROC-ESM-CHEM | GFDL-ESM2G | IPSL-CM5A-MR | CSIRO-Mk3-6-0 |
| MIROC5 | GFDL-ESM2M | IPSL-CM5B-LR | **FGOALS-s2** |
| NorESM1-M | MPI-ESM-LR | MRI-CGCM3 | **GFDL-CM3** |
| NorESM1-ME | MPI-ESM-MR | MRI-ESM1 | HadGEM2-AO |
| | MPI-ESM-P | | HadGEM2-CC |
| | | | HadGEM2-ES |


| ZModel-group 1 | ZModel-group 2 | ZModel-group 3 | ZModel-group 4 |
|---|---|---|---|
| ACCESS1-0 | **CMCC-CMS** | BNU-ESM | ACCESS1-3 |
| bcc-csm1-1-m | **CNRM-CM5** | CanCM4 | bcc-csm1-1 |
| CCSM4 | **CNRM-CM5-2** | CanESM2 | CSIRO-Mk3-6-0 |
| CESM1-BGC | **FGOALS-s2** | CMCC-CM | HadGEM2-AO |
| CESM1-CAM5 | **GFDL-CM3** | FGOALS-g2 | HadGEM2-CC |
| CESM1-CAM5-1-FV2 | | IPSL-CM5A-LR | HadGEM2-ES |
| CESM1-FASTCHEM | | IPSL-CM5A-MR | MIROC-ESM |
| CESM1-WACCM | | MRI-CGCM3 | MIROC-ESM-CHEM |
| GISS-E2-H | | NorESM1-M | MRI-ESM1 |
| GISS-E2-H-CC | | NorESM1-ME | |
| GISS-E2-R | | | |
| GISS-E2-R-CC | | | **ZModel-group 5** |
| HadCM3 | | | |
| inmcm4 | | | CMCC-CESM |
| IPSL-CM5B-LR | | | GFDL-CM2p1 |
| MIROC5 | | | GFDL-ESM2G |
| MPI-ESM-LR | | | GFDL-ESM2M |
| MPI-ESM-MR | | | |
| MPI-ESM-P | | | |


Table A1: Composition of the different Model-groups identified in the main text. In bold, we
show the CMIP5 models which belong to Model-group 4 and ZModel-group 2. We note that all
the models belonging to Zmodel-group 2 also belong to Model-group 4.





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





| nb | Model Acronym | nb | Model Acronym |
|----|---------------|----|---------------|
| 1 | bcc-csm1-1 | 25 | HadGEM2-ES |
| 2 | bcc-csm1-1-m | 26 | MPI-ESM-LR |
| 3 | BNU-ESM | 27 | MPI-ESM-MR |
| 4 | CanCM4 | 28 | MPI-ESM-P |
| 5 | CanESM2 | 29 | MRI-CGCM3 |
| 6 | CMCC-CESM | 30 | MRI-ESM1 |
| 7 | CMCC-CM | 31 | GISS-E2-H |
| 8 | CMCC-CMS | 32 | GISS-E2-H-CC |
| 9 | CNRM-CM5 | 33 | GISS-E2-R |
| 10 | CNRM-CM5-2 | 34 | GISS-E2-R-CC |
| 11 | ACCESS1-0 | 35 | CCSM4 |
| 12 | ACCESS1-3 | 36 | NorESM1-M |
| 13 | CSIRO-Mk3-6-0 | 37 | NorESM1-ME |
| 14 | inmcm4 | 38 | HadGEM2-AO |
| 15 | IPSL-CM5A-LR | 39 | GFDL-CM2p1 |
| 16 | IPSL-CM5A-MR | 40 | GFDL-CM3 |
| 17 | IPSL-CM5B-LR | 41 | GFDL-ESM2G |
| 18 | FGOALS-g2 | 42 | GFDL-ESM2M |
| 19 | FGOALS-s2 | 43 | CESM1-BGC |
| 20 | MIROC-ESM | 44 | CESM1-CAM5 |
| 21 | MIROC-ESM-CHEM | 45 | CESM1-CAM5-1-FV2 |
| 22 | MIROC5 | 46 | CESM1-FASTCHEM |
| 23 | HadCM3 | 47 | CESM1-WACCM |
| 24 | HadGEM2-CC | | |


Table 1: List of the CMIP5 models used for the comparison. The reader is referred to the CMIP5
documentation for more information on each of them. Here, each configuration is furthermore
given a number, for easier identification in subsequent figures.





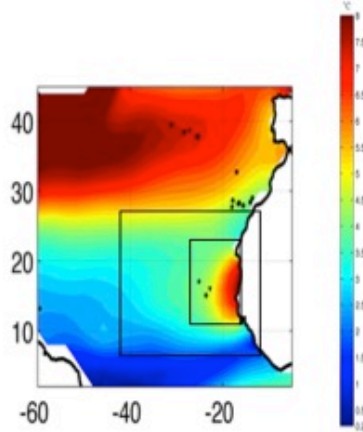

Figure 1: Amplitude of the SST seasonal cycle in the western tropical north Atlantic. SST data are from the ERSSTv3b data set averaged between 1975 and 2005. The two black boxes show the extended and zoomed regions respectively over which the statistical classifications were performed (see text for details).



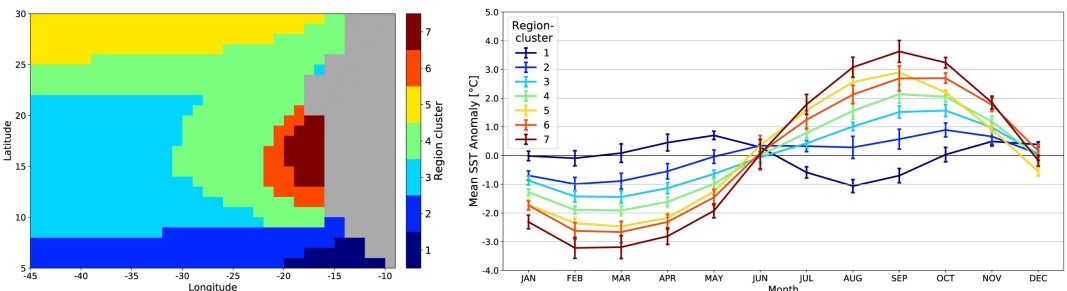

760

Figure 2: Left panel: Region-clusters associated with the SOM-clusters obtained after a HAC on
a 30x4 neuron SOM learned on ERSSTv3b observations in the extended zone (see text for
details). Right Panel: Ensemble-mean climatological SST cycles for the grid points of the seven
Region-clusters. The error bars show the standard deviation of this ensemble mean.

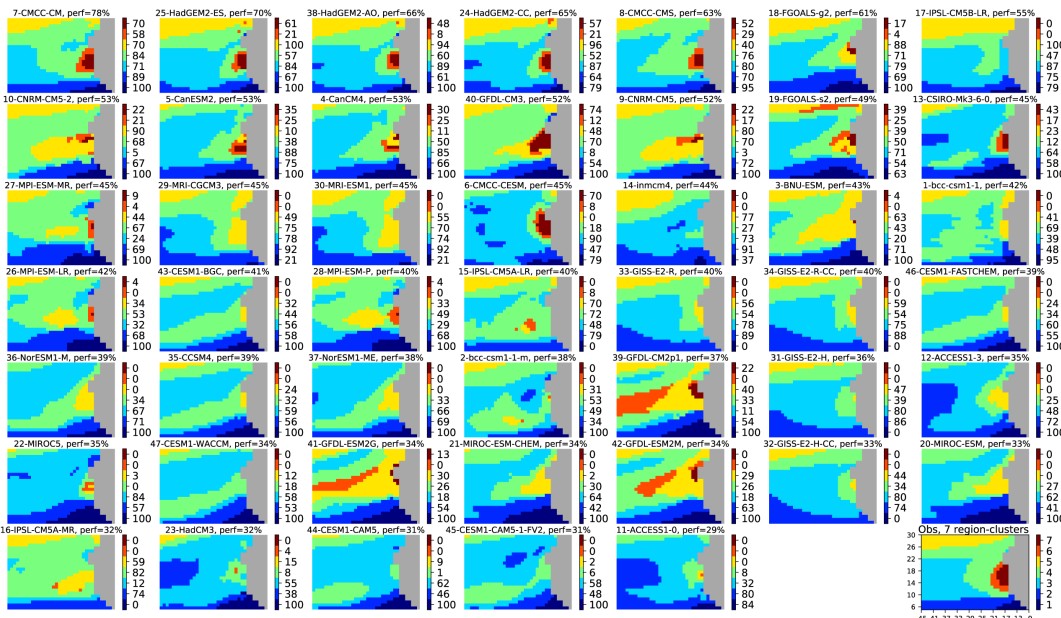

Figure 3: Projection of the 47 climate models of the CMIP5 database onto the SOM learned with ERSSTv3b climatology in the extended zone (see Fig. 1). On top of each panel, we figure: the number referencing the model, its name (Table 1), and its skill given as a mean percentage (see text). The models are ordered according to their skill in decreasing order. The 7 Region-clusters (or SOM-clusters) are defined by applying an HAC to the SOM output learned with the observation field. They are represented by different colors. The numbers in the colorbar at the right of each panel represent the skill for each Region-cluster. The observation field is shown in the bottom right panel and the numbers in front of the colorbar reference the Region-cluster.



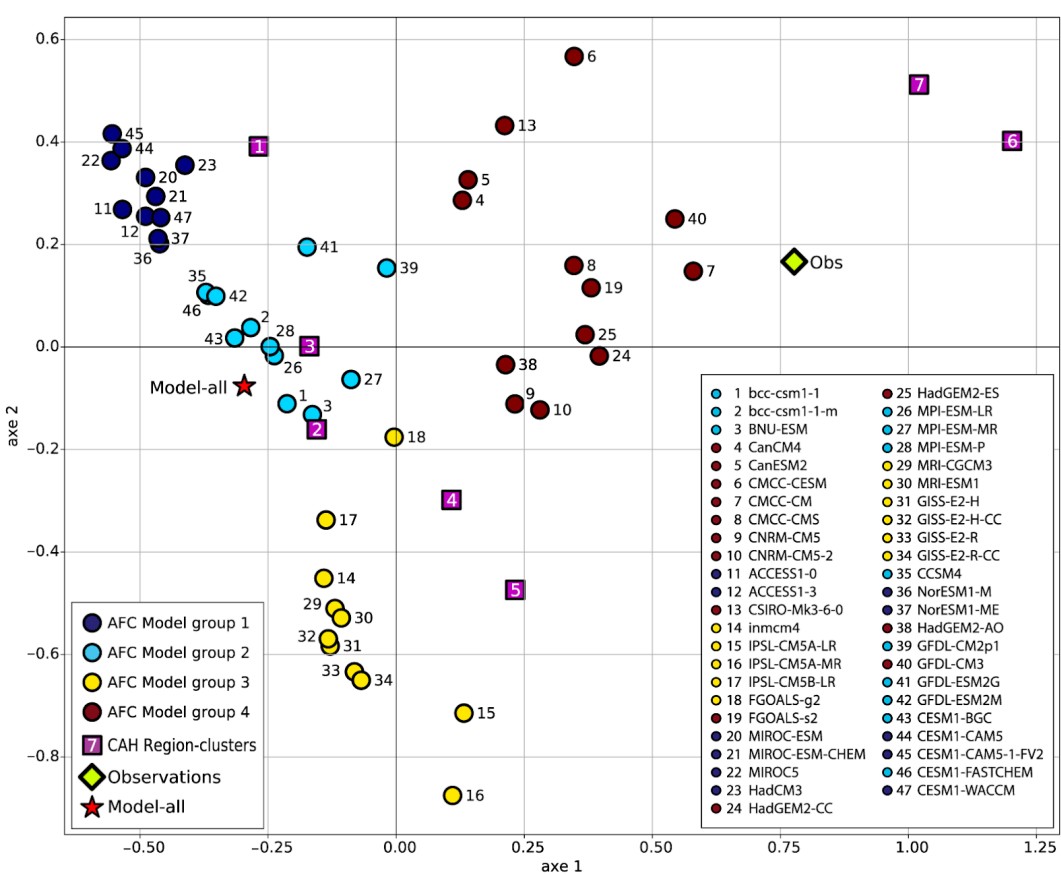


Figure 4: Projection of the CMIP5 models (colored circles) and the observation field (green
diamond) defined by their cluster skill vectors on the first two axis of the MCA. The seven
region-clusters of the observation field are represented by purple squares. The colours of the
circles denote the four groups of models obtained after an HAC was performed on the seven
MCA components of the models. The projection of the full multi-model mean (47 models) is
represented by a red star.





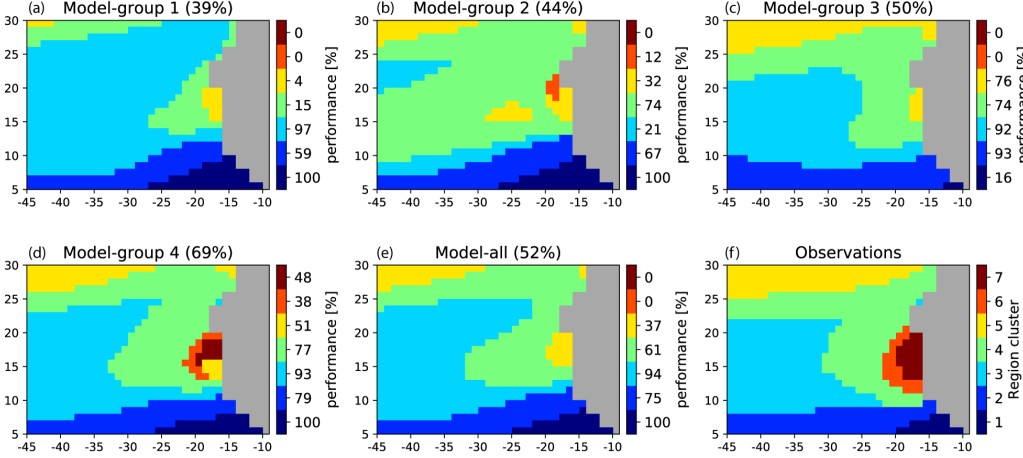


Figure 5: (a)-(d): Projection of the multi-model ensembles (Model-group) onto the SOM learned
with ERSSTv3b climatology in the extended zone. Multi-model ensemble performances are
obtained by averaging the skill of the models forming each group. The performances are given
on top of each panel. The Region-clusters determined by processing the observations in the
extended area and their associated colors are given in the bottom right panel. The colorbars at the
right of each multi-ensemble panel represent the skill (in %) associated with each Region-cluster.
Panel (e) shows the same for the full multi-model ensemble. Panel (f) reproduces the Region-
clusters based on the observations also shown in Fig. 2.





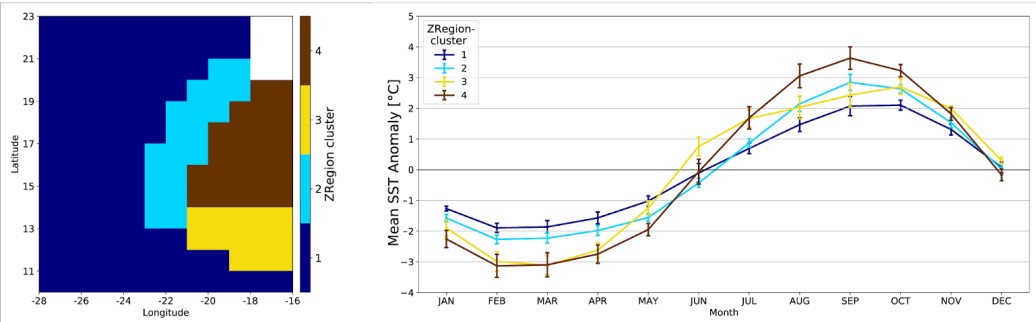


Figure 6: Left panel: ZRegion-clusters associated with the ZSOM-clusters obtained after a HAC
on a 10x12 neuron SOM learned on ERSSTv3b observations in the zoomed zone (see text for
details). Right Panel: Ensemble-mean climatological SST cycles for the grid points of the four
ZRegion-clusters. The error bars show the standard deviation of this ensemble mean.





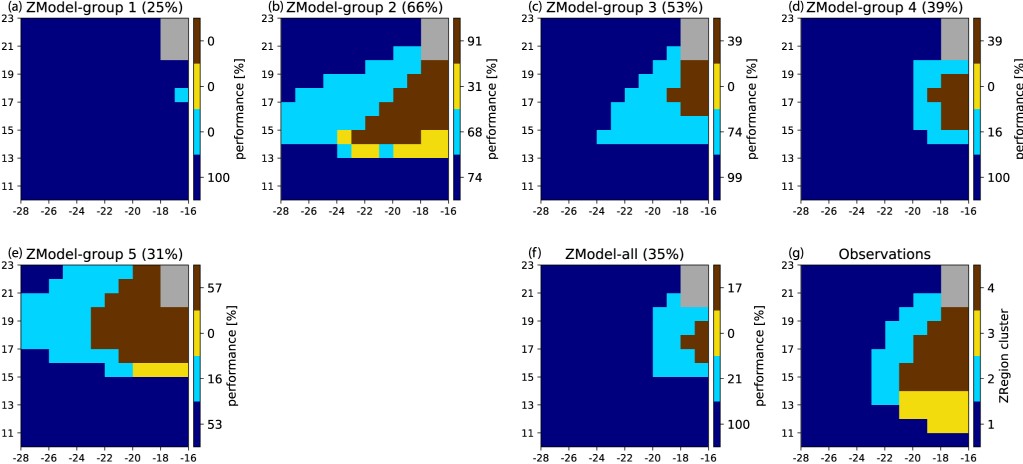


Figure 7: (a)-(e): Projection of the multi-model ensembles (ZModel-groups) onto the ZSOM. The
performances are given on top of each panel. The ZRegion-clusters determined by processing the
observations in the zoomed region and their associated colors are given in the bottom right panel.
The colorbars at the right of each multi-ensemble panel represent the skill (in %) associated with
each ZRegion-cluster. Panel (f) shows the same for the full multi-model ensemble. Panel (g)
reproduces the Region-clusters based on the observations also shown in Fig. 6.

811



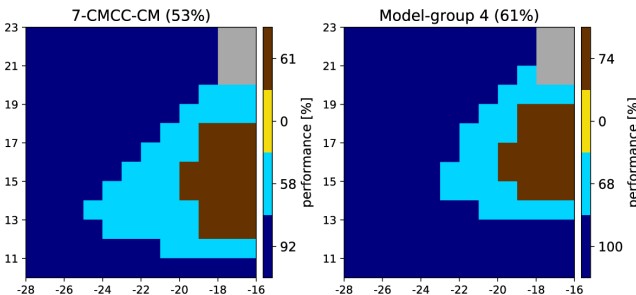

812

Figure 8 : Same as Fig. 7 but for the individual model CMCC-CM (model 7) (left) and the
Model-group 4 (right).

815



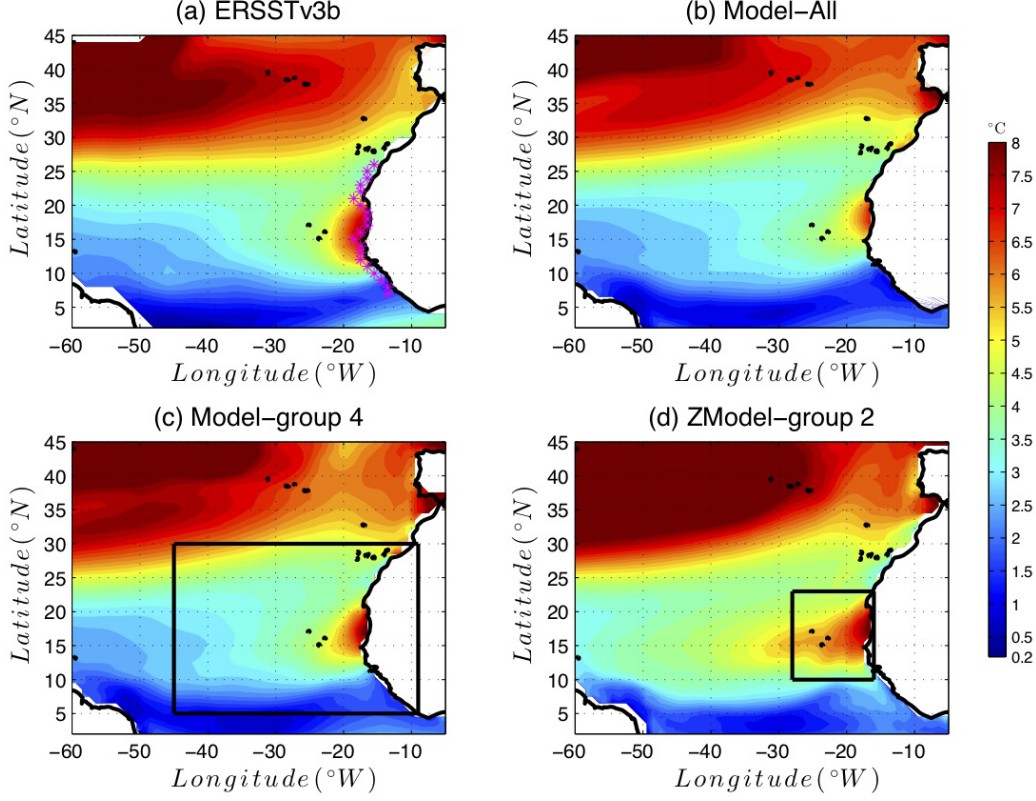

Figure 9: Amplitude of the SST seasonal cycle in the (a) ERSSTv3b Observations (b) Model-All,
c) Model-group 4 (best Model-group for the exended area, figured out by the black rectangular
box) and (d) ZModel-group 2 (best Model-group for the reduced area, figured out by the small
black rectangular box). The SST seasonal cycle is computed over the period 1985-2005



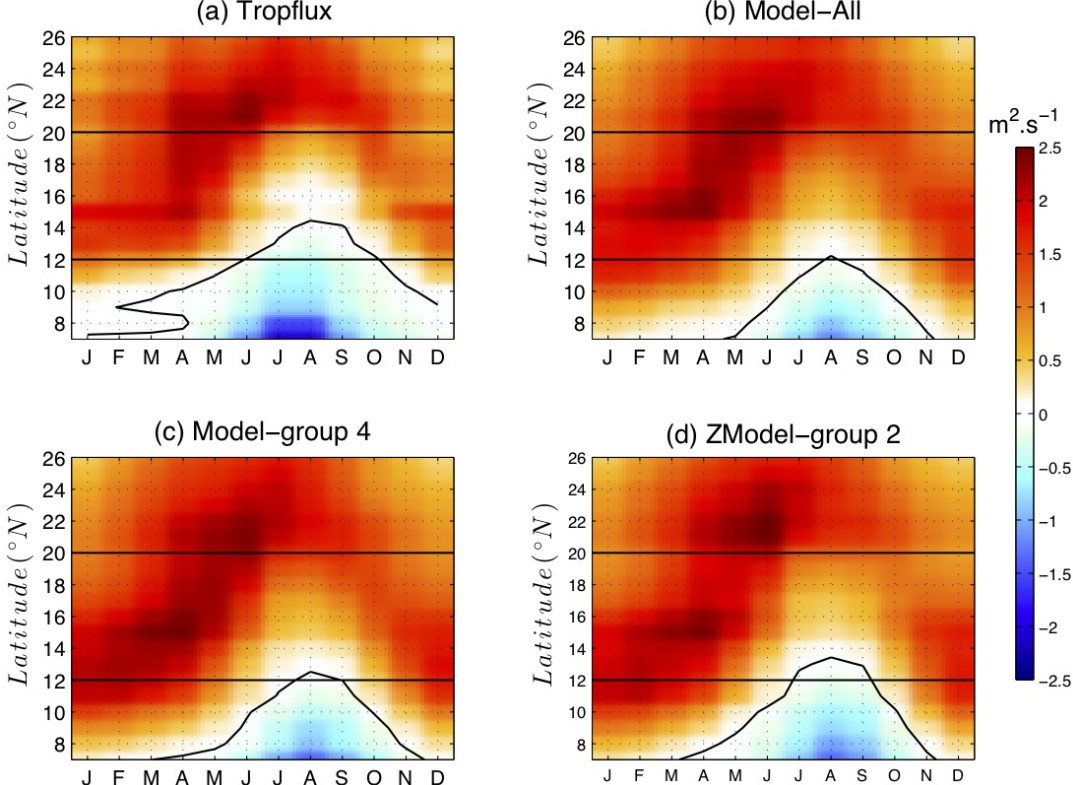

822

Figure 10: Latitude-time plot of depth integrated Ekman transport computed over the grid point located along the coast (magenta stars in Fig. 9.a). The time axis shows climatological months over the period 1985-2005. Positive (negative) values correspond to upwelling (downwelling) conditions. Panel (a) stands for TropFlux data set (see Praveen Kumar et al. (2013) (b) Model-All, (c) Model-group 4 and (d) ZModel-group 2. On each panel, the black contour shows the contour zero. The horizontal dashed lines are positioned at 12°N and 20°N and give a rough limitation of the senegalo-mauritanian upwelling region.



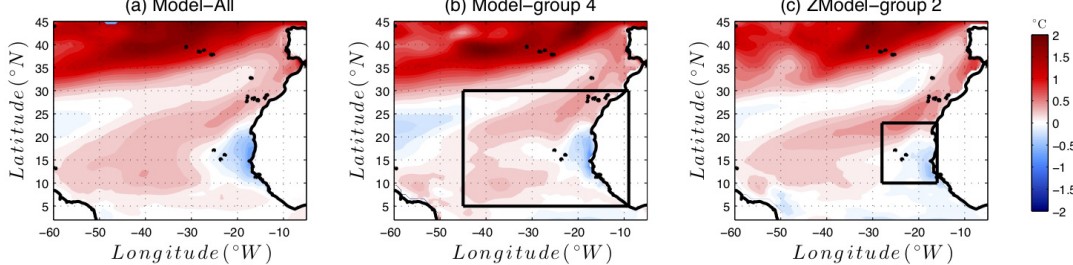

832

Figure 11: Evolution of the amplitude of the SST seasonal cycle at the end of the 21[st] century.
The figure shows the difference between the seasonal cycle amplitude averaged over the period
[2080-2100] following the RCP8.5 scenario and the amplitude averaged over the period [1985-
2005] in the historical simulations. A positive value (red) means that the seasonal cycle is more
marked over the period 2080-2100.

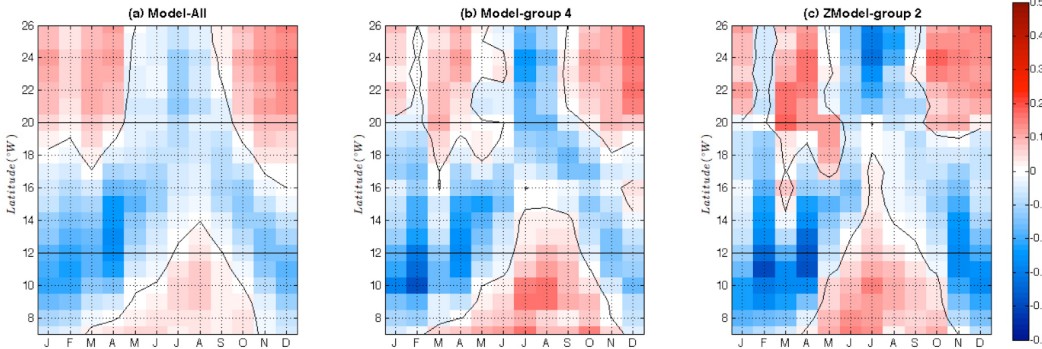

840

Figure 12: Latitude-time diagram of the seasonal shift of the meridional component of the wind-stress with respect to the present days. For each month and at each latitude, we show the meridional wind stress shift with respect to the present days averaged over the period [2080-2100]. Positive values (red) means that the wind stress shift is southward and is thus favorable to upwelling. Panel (a) stands for Model-All, (b) Model-group 4 and (c) ZModel-group 2.

846