# Peer review of "Towards an objective assessment of climate multi-model"

_Geoscientific Model Development, 2019_

## Referee Comment (RC1) · Anonymous Referee #1 · 23 Feb 2020

General comments

In this paper, the authors develop a statistical method for assessing CMIP5 climate model simulations of upwelling in the Senegalo-Mauritanian upwelling region and briefly discuss future projections of upwelling from a subset of the best-performing models. The method for assessing the models appears sound and seems to produce acceptable results in evaluating the models. However, I found the description of the method and its application difficult to follow at some points, as detailed in the specific comments below. There are also a number of typographical, grammatical, and organizational issues which impede the reader's ability to interpret the writing at some points.

[Figure]

I have provided some specific comments on some corrections needed in the technical corrections section, but this is not an exhaustive list and the authors should carefully proofread the paper prior to submitted any revised version. Finally, there seems to be a mismatch in the wind data discussed in the Data section versus the wind data used to produce Figure 10, as detailed in the specific comments below. All of these issues should be corrected before any subsequent version of the paper can be evaluated.

Specific comments

Lines 22-23: Give a brief summary of the main findings on the future behavior of the Senegalo-Mauritanian upwelling in the abstract, rather than simply stating that the future behavior was assessed.

Lines 106-107: More detail is needed on the ERSST_v3b data set. How is this dataset produced, and why was it chosen as the "observation field" for comparison with the models?

Lines 112-114: Similar to the ERSST_v3b data set, please provide some additional detail on the QUICKSCAT (sic) product. And is this product actually used in the paper? In the caption for figure 10, the TropFlux data set is referenced rather than QuikSCAT in the discussion of the wind stress and Ekman transport (see comment on line 826), and there is no mention of QuickSCAT or TropFlux in the discussion of this figure in section 5.1. Additionally, "QuikSCAT" is the correct spelling of this satellite.

Line 118: Does "Sylla et al. (in rev.)" refer to the Sylla et al. 2019 Climate Dynamics paper, or another work? If this is another paper, it should be added to the reference list.

Section 3.2 (starting on line 168): It's not clear to me why the SOM classification followed by the HAC clustering was necessary. What is the reason for performing both classifications rather than just using one method or the other?

Lines 197-198: What is a "standard statistic algorithm"? Is this referring to the calculation of the standard deviation?

Section 3.4 (lines 255-330): Perhaps it is just my own ignorance, but I find figure 4 and the accompanying discussion quite difficult to interpret. What, conceptually, do the x and y axes and the grouping of the points on the plot represent? The description says that "proximity between a model and a region-cluster leads us to affirm that this region-cluster is well represented by that model", but the observations and the "highest skill" models 7 and 25 are far away from any of the region clusters...? And I think I understand that model 7 is considered as having good skill because it lies close to the "obs" point on the plot, but why is model 25 considered to have better skill than models 24, 19, 8, and 40, which are located a similar distance from the "obs" point as model 7? Have any previous studies used this method to assess the skill of models?

Line 826: What is the TropFlux data set? This needs to be described and its use justified in the Data section.

Technical corrections

The writing in this paper is frequently conversational in tone rather than technical, and there are many instances of imprecise filler words like "very", "pretty", and "nicely". In line 250, "let us say..." is a conversational phrase that is not appropriate to use in a scientific manuscript in this context. Also, the mention of "ongoing studies in our group" (line 527) is fine for a conference presentation but, in my opinion, is not appropriate to write in a scientific paper. Please proofread the paper and correct these and other such instances of informal language.

There are several excessively long paragraphs that are taxing on the reader. For example, the paragraph from lines 30-73 in the Introduction and the paragraph from lines 332-377 in section 4 are very difficult to read due to their length. Please break up and reorganize these and other long paragraphs.

There are a number of typographical and grammatical errors throughout the paper,

which impede its interpretability to the reader in some places. I have given a few examples below, but this is not an exhaustive list, and the authors should check the entire paper carefully for such errors in any subsequent versions of the manuscript.

- Title: Extra space after the word "in"

- I am not able to read the full "short summary" on the discussion paper web site, but the part I can see contains three misspelled words.

- Errors in capitalization of words (e.g. "Observation field" in line 108, seasonal "Cycle" in line 299)

- Lines 16-18: Abrupt shift from first-person to third person ("We used a neural classifier…" to "One can then determine…")

- Line 83: Typo ("lies at is the southern…")

- Lines 109-110: Typo ("were been regridded")

- Line 525: Typo ("costal" instead of "coastal")

Line 16: Typically self-organizing maps are described as an "artificial neural network" rather than a "neural classifier".

Line 21: What is meant by the phrase "performing multi-model ensemble"?

Line 26: CMIP5 stands for the "Coupled Model Intercomparison Project, Phase 5" (not the "5th Climate Model Intercomparison Project").

Line 383: Shouldn't this say "(Fig. 8, left)"?

Line 755: Figure 1 appears distorted and blurry in the PDF version of the paper. Please correct this figure to make it easier to read.

---

## Referee Comment (RC2) · Anonymous Referee #2 · 3 Mar 2020

Mignot et al. use self organising maps to evaluate the behaviour of a multi-model ensemble in the Senegalo-Mauritanian upwelling region, with the aim of producing accurate projections of future climate changes in the region. Their algorithm aims to select models that yield a specific desired quantity - in this case, a multi model mean. They then project the selected models through the future to assess changes in the region.

There is clearly a great deal of potential in the technical work in this paper. The idea of using Self Organising Maps as a dimension reduction and interpretation technique is a good one, and appears to work very well. It can clearly add a great deal of value

to the analysis of a large multi-model ensemble in this region. However, I feel a degree of restructuring, clarification of the aims of the paper, and editing for overall clarity is required before the scientific content can be properly assessed.

I feel the paper would most benefit from restructuring so that the objectives, details and then method of assessment of the algorithm were more clearly laid out earlier in the paper, and the reader were more carefully led through that process. As it stands, intense technical detail follows very broad overview statements, and important details about the analysis are left until later in the paper, so the reader is left confused and searching for appropriate context into which to place technical detail. Some choices in the analysis feel arbitrary, and it is unclear whether this is because they are indeed arbitrary, or that they are inadequately described.

The most obvious candidate for restructuring is the start of section 3, describing the methods used for classification of the models. This section dives straight into a detailed description of self organising maps (SOMs), without a discussion of precisely what the algorithm aims to achieve, how that can be assessed, and why SOMs were chosen as opposed to any other dimension reduction technique. As it is, the paper reads as "we decided to use SOMS and this is what you can do with them", rather than "we are trying to solve a specific problem and here is how SOMs can help". One suggestion would be to take the description of the methods from the start of section 6 (Discussion and Conclusion), expand upon it and place it at the start of section 3.

The SOMs appear to successfully cluster the model field into regions with different dynamics. This seems useful and interesting. How does it help solve the specific problem? I think it would be useful to set out near the beginning of the paper the exact strategy that will be used, and how to tell if it is successful or not. For example, It seems clear that the assessment algorithm (starting line 232) can be used to rank the models in terms of their closeness to observations and dynamics in particular regions. One downside however, is that it does not give the modeller an intuition into how far the model is from "good" behaviour in absolute terms. We simply get an averaged "skill

score" from 28% to 79%, but without an idea of how this might relate to more traditional measures of skill. So how close is the best model and how far is the worst model from reality? We have only a score (useful as that is) to guide us.

The paper makes the claim that it offers an objective method for the assessment of the behaviour of models with regards historical observations. I struggle to accept this, given the number of subjective choices made with regards to the way the analysis is conducted. Subjective judgements will always need to be made in the analysis of climate model output - this is inevitable, and perfectly reasonable as long as labelled as such. The paper only examines a subset of model fields for example, and a subjective choice as to which of those fields to select has been made.

A core problem that needs to be addressed in the paper can be illustrated by considering the section starting on line 285:

"As indicated in the introduction, the main objective of the methodology is to select an ensemble of models that represents at best the upwelling behavior with respect to the observations and to use this ensemble to predict the impact of climate change in the Senegalo-Mauritanian upwelling with some confidence. The problem is now to determine a subset of models that can adequately represent the observations, as the number of models is small enough we choose to cluster them by HAC according to their projections onto the seven axes provided by the MCA, and select the optimal jump in the hierarchical tree (Jain and Dubes, 1998)."

I cannot see a description of what it means for a subset of models to "adequately represent the observations". I also cannot see an adequate description for what the "optimal jump in the heirarchical tree" of Jain and Dubes (1998) is, or what it might mean for the ensemble members. The clustering of the models in figure 4 looks reasonable by eye, but there are a large number of other ways that the models could be clustered that might be equally as reasonable.

The authors claim that their algorithm selects a number of ensemble members that best

represent an ensemble mean. I don't believe that they provide sufficient justification for why the ensemble mean should be selected for, or that the ensemble members their algorithm selects members in a way that is superior to a subjective selection. This is presented as a "model weighting" paper, and while that might be possible with this algorithm, I do not believe that is where the strength of the analysis lies. The paper would be better re-cast as a model analysis paper, using an interesting and useful algorithm to explore the dynamical deficiencies of the models in the region, and informing climate modellers of those deficiencies. I think if the authors wish it to be a model weighting paper, then more emphasis needs to be given to the meaning and justification of the weighting scheme. Further, the authors should develop placing the weighting scheme in the context of established work on the meaning of multi-model ensembles.
* * *

---

## Author Comment (AC1) · 15 Apr 2020

**Reviewer 1**

General comments
In this paper, the authors develop a statistical method for assessing CMIP5 climate model simulations of upwelling in the Senegalo-Mauritanian upwelling region and briefly discuss future projections of upwelling from a subset of the best-performing models. The method for assessing the models appears sound and seems to produce acceptable results in evaluating the models. However, I found the description of the method and its application difficult to follow at some points, as detailed in the specific comments below. There are also a number of typographical, grammatical, and organizational issues which impede the reader's ability to interpret the writing at some points.

I have provided some specific comments on some corrections needed in the technical corrections section, but this is not an exhaustive list and the authors should carefully proofread the paper prior to submitted any revised version. Finally, there seems to be a mismatch in the wind data discussed in the Data section versus the wind data used to produce Figure 10, as detailed in the specific comments below. All of these issues should be corrected before any subsequent version of the paper can be evaluated.

We would like to thank the reviewer for his/her careful reading of the manuscript and his/her relevant comments. We answered them carefully, and we believe the manuscript has definitely improved. We apologize for the typos and grammatical issues underlined by the reviewer. We have carefully revised the whole manuscript to improve this point specifically. We detail below ll the modifications that have been implemented in the text.

Specific comments
Lines 22-23: Give a brief summary of the main findings on the future behavior of the Senegalo-Mauritanian upwelling in the abstract, rather than simply stating that the future behavior was assessed.
This sentence was changed into "The future reduction of the Senegalo-Mauritanian upwelling proposed in recent studies is then revisited using this multi-model selection."

Lines 106-107: More detail is needed on the ERSST_v3b data set. How is this dataset produced, and why was it chosen as the "observation field" for comparison with the Models?
Some details on the ERSST_v3b were added. We also explained that other data sets would have been available, in particular HadISST. A previous study has shown that differences in the Senegal-Mauritania upwelling are relatively weak. Yet, sensitivity of the method to the target field should definitely be addressed in the future, we thank the reviewer for this remark.

Lines 112-114: Similar to the ERSST_v3b data set, please provide some additional detail on the QUICKSCAT (sic) product. And is this product actually used in the paper?

In the caption for figure 10, the TropFlux data set is referenced rather than QuikSCAT in the discussion of the wind stress and Ekman transport (see comment on line 826), and there is no mention of QuickSCAT or TropFlux in the discussion of this figure in section 5.1. Additionally, "QuikSCAT" is the correct spelling of this satellite.

We apologize for the typo on the name of the satellite. We also apologize for the fact that QuickSCAT data is indeed not used in the final version of the paper. The wind stress was evaluated against the TropFlux reanalysis. Information about this product was added (end of section 2.1)

Line 118: Does "Sylla et al. (in rev.)" refer to the Sylla et al. 2019 Climate Dynamics paper, or another work? If this is another paper, it should be added to the reference List.

Sylla et al. (in rev.) and Sylla et al (2019) are indeed the same paper, accepted in the course of the preparation of the present manuscript. The citations have been homogenized, we apologize for this.

Section 3.2 (starting on line 168): It's not clear to me why the SOM classification followed by the HAC clustering was necessary. What is the reason for performing both classifications rather than just using one method or the other?

The SOM model has been used to determine a vector quantization of the dataset: i.e. to determine referent vectors that are a representative summary of the learning dataset. The vector quantization compresses the total database into a quite small (with respect to the size of the database) number of referent vectors such as each data is not too different of its nearest referent according to a distance (The Euclidean distance in the present case). The exact number of referents, that is the number of neurons, does not really matter because this number will be reduced by the HAC. Doing so allows us to take the non-linearities of the dataset into account in the analysis. The exact number at the end of the SOM+HAC procedure is not known a priori but at the end of the study by looking at the HAC dendrogram, which suggests several possibilities for the number of classes to estimate. A compromise between the number of classes we can explain from a physical point of view and the number we need to include the information embedded in the dataset is made. This procedure has been used with success in several papers (Jouini et al, 2016, JGR; Farikou et al, 2015, JGR; Sawadogo et al, 2009, IEEE; Niang et al, 2003, RSE).

We have added an introduction at the beginning of section 3 to explain this point. Note that it was also raised by reviewer 2. We are grateful to both reviewers for requiring this classification.

Lines 197-198: What is a "standard statistic algorithm"? Is this referring to the calculation of the standard deviation?

It refers indeed to the calculation of the spread in each neuron. We have re-phrased this sentence as:

"for each region-cluster, we estimated the monthly mean of the SST seasonal cycles and the associated spread captured by the neurons constituting this region-cluster" . Thank you for this clarification.

Section 3.4 (lines 255-330): Perhaps it is just my own ignorance, but I find figure 4 and the accompanying discussion quite difficult to interpret. What, conceptually, do the x and y axes and the grouping of the points on the plot represent? The description says that "proximity between a model and a region-cluster leads us to affirm that this region-cluster is well represented by that model", but the observations and the "highest skill" models 7 and 25 are far away from any of the region clusters. . .? And I think I understand that model 7 is considered as having good skill because it lies close to the "obs" point on the plot, but why is model 25 considered to have better skill than models 24, 19, 8, and 40, which are located a similar distance from the "obs" point as model 7? Have any previous studies used this method to assess the skill of models?

Sub-section 3.4 is now section 4. In fact the MCA method used in this section is different from the SOM and quite new in geophysics; it therefore deserves a dedicated section in which we give more details on the functioning of the MCA. We have rewritten the presentation of the MCA analysis in order to get an easier understanding of the functioning of this analysis and to facilitate the understanding of Figure 4. The new writing of the beginning of the MCA presentation (section 4) is shown below. Our changes are in yellow in this new writing.

"In order to further progress in the selection of the models, the 47 climate models and the Observation field were then analyzed by using a Multiple Correspondence Analysis (MCA in the following). MCA is a multivariate statistical technique that is conceptually similar to principal component analysis (PCA in the following), but applies to categorical rather than continuous data. Similarly as PCA, it provides a way of displaying a set of data in a two-dimensional graphical form.

In the following, we applied a MCA analysis to the (47 , 7) matrix $\mathbf{R}$ = [$Rmi$] whose elements represent the skills of the clusters of the models shown in front of the color bars in Fig. 3: the rows m represent the 47 different models, the columns $i$ the 7 region-clusters. The MCA, as the PCA does, projects the initial matrix in a new basis in such a way that the new axes are the matrix eigenvectors (PC), the inertia of each axe being the related eigenvalues. According to the theory, the MCA matrix analysis gives 6=(7-1) independent PCs. Each model is now associated with a 6-dimensional vector. The MCA uses for this analysis the khi 2 distance. In figure 4, we present the projection of the models and the "region clusters" in the plane formed by the two first axes (x=PC1 and y= PC2) of the MCA that represent 70 % of the total inertia. Each model is represented by a small circle and each Region-cluster by a purple square. Moreover, we projected the observation field (green diamond) on that plane as a supplementary individual. The proximities in figure 4 is represented by the khi2 distance. To have a more precise view, it should be necessary to consider the projection on the 5 other PCs which represent 30% of the inertia.

In the (PC1, PC2) plane, the shorter the distance between two models, the more similar the distribution of their region-cluster skills. The seven clusters of the observation field are represented by purple squares. Proximity between a model and a region-cluster leads us to affirm that this region-cluster is well represented by that model. ..;

……………

……………;

   In this above analysis, we must be aware that we are confronted to the intrinsic difficulty to represent multidimensional data in a plan. The representation of some data can be biased thank to the importance given to the other axes."

Line 826: What is the TropFlux data set? This needs to be described and its use justified in the Data section.

The TropFlux reanalysis combines the ERA-Interim reanalysis for turbulent and long-wave fluxes, and ISCCP (International Satellite Cloud Climatology Project) surface radiation data for shortwave fluxes. This wind stress product is described and evaluated in Praveen Kumar et al. (2011).

These lines were added in the Data section.

Technical corrections

The writing in this paper is frequently conversational in tone rather than technical, and there are many instances of imprecise filler words like "very", "pretty", and "nicely".

In line 250, "let us say. . ." is a conversational phrase that is not appropriate to use in a scientific manuscript in this context.

Also, the mention of "ongoing studies in our group" (line 527) is fine for a conference presentation but, in my opinion, is not appropriate to write in a scientific paper.

Please proofread the paper and correct these and other such instances of informal language.

The text was largely revised and improved in this respect. The two specific sentences cited above were corrected. We thank the reviewer for this remark that led us to significantly improve the language of the manuscript.

There are several excessively long paragraphs that are taxing on the reader. For example, the paragraph from lines 30-73 in the Introduction and the paragraph from lines 332-377 in section 4 are very difficult to read due to their length. Please break up and reorganize these and other long paragraphs.

These two paragraphs and several others have been cut. Thanks also for this remark that improves the readability of the paper.

There are a number of typographical and grammatical errors throughout the paper, which impede its interpretability to the reader in some places. I have given a few examples below, but this is not an exhaustive list, and the authors should check the entire paper carefully for such errors in any subsequent versions of the manuscript.

- Title: Extra space after the word "in"

Corrected

- I am not able to read the full "short summary" on the discussion paper web site, but the part I can see contains three misspelled words.

We are not sure why the short summary could not be read from the website, but the short summary was corrected

- Errors in capitalization of words (e.g. "Observation field" in line 108, seasonal "Cycle" in line 299)
These capitalizations intended to put emphasis on the elements of the mathematical procedure. Yet, we realize that this was not clear so that we removed all the capital letters in these expressions.

- Lines 16-18: Abrupt shift from first-person to third person ("We used a neural classifier. . ." to "One can then determine. . .")
corrected

- Line 83: Typo ("lies at is the southern. . .")
corrected

- Lines 109-110: Typo ("were been regridded")
corrected

- Line 525: Typo ("costal" instead of "coastal")
corrected

Line 16: Typically self-organizing maps are described as an "artificial neural network" rather than a "neural classifier".
This was changed: In the text we now use a artificial neural network (Self-Organizing Maps),

Line 21: What is meant by the phrase "performing multi-model ensemble"?
This phrase was changed to "an efficient multi-model combination of 12 climate models"

Line 26: CMIP5 stands for the "Coupled Model Intercomparison Project, Phase 5" (not the "5th Climate Model Intercomparison Project").
corrected

Line 383: Shouldn't this say "(Fig. 8, left)"?
Corrected, apologizes for this mistake

Line 755: Figure 1 appears distorted and blurry in the PDF version of the paper. Please correct this figure to make it easier to read.
This figure was corrected and the .eps version attached to the revisions is not blurred.

---

## Author Comment (AC2) · 15 Apr 2020

**Reviewer 2**

Mignot et al. use self organising maps to evaluate the behaviour of a multi-model ensemble in the Senegalo-Mauritanian upwelling region, with the aim of producing accurate projections of future climate changes in the region. Their algorithm aims to select models that yield a specific desired quantity - in this case, a multi model mean. They then project the selected models through the future to assess changes in the region.

There is clearly a great deal of potential in the technical work in this paper. The idea of using Self Organising Maps as a dimension reduction and interpretation technique is a good one, and appears to work very well. It can clearly add a great deal of value to the analysis of a large multi-model ensemble in this region. However, I feel a degree of restructuring, clarification of the aims of the paper, and editing for overall clarity is required before the scientific content can be properly assessed.

We would like to thank the reviewer for his/her careful reading of the manuscript and his/her constructive and challenging comments. We have restructured the manuscript as suggested, and clarified the methodology as much as we could. This has greatly improved the manuscript. We have also paid specific attention to editing issues for which we apologize. We detail below on all the modifications that have been implemented the text.

I feel the paper would most benefit from restructuring so that the objectives, details and then method of assessment of the algorithm were more clearly laid out earlier in the paper, and the reader were more carefully led through that process. As it stands, intense technical detail follows very broad overview statements, and important details about the analysis are left until later in the paper, so the reader is left confused and searching for appropriate context into which to place technical detail. Some choices in the analysis feel arbitrary, and it is unclear whether this is because they are indeed arbitrary, or that they are inadequately described.

The most obvious candidate for restructuring is the start of section 3, describing the methods used for classification of the models. This section dives straight into a detailed description of self organising maps (SOMs), without a discussion of precisely what the algorithm aims to achieve, how that can be assessed, and why SOMs were chosen as opposed to any other dimension reduction technique. As it is, the paper reads as "we decided to use SOMS and this is what you can do with them", rather than "we are trying to solve a specific problem and here is how SOMs can help". One suggestion would be to take the description of the methods from the start of section 6 (Discussion and Conclusion), expand upon it and place it at the start of section 3.

The SOMs appear to successfully cluster the model field into regions with different dynamics. This seems useful and interesting. How does it help solve the specific problem? I think it would be useful to set out near the beginning of the paper the exact strategy that will be used, and how to tell if it is successful or not.

Thank you for these detailed and constructive remarks. We restructured the paper by taking the remarks of the reviewer exposed in the above paragraphs into account. First, at the end of the introduction we now mention that our method is based on classification. At the beginning of section 3, we now justify the use of a classification method on the one hand and then choice of the SOM+HAC on the other hand.

Moreover, Sub-section 3.4 is now section 4. In fact the MCA method used in this section is different from the SOM and quite new in geophysics; it therefore deserves a dedicated section in which we give more details on the functioning of the MCA.

We also cut some long paragraphs in smaller paragraphs in order to facilitate the understanding of the text.

Regarding the relevance of the SOM in particular, note that this question was also raised by reviewer 1. We are grateful to both reviewers for requiring this classification. As answered above, the SOM model has been used to determine a vector quantization of the dataset: i.e. to determine referent vectors that are a representative summary of the learning dataset. The vector quantization compresses the total database into a quite small (with respect to the size of the database) number of referent vectors such as each data is not too different of its nearest referent according to a distance (The Euclidean distance in the present case). The exact number of referents, that is the number of neurons, does not really matter because this number will be reduced by the HAC. Doing so allows us to take the non-linearities of the dataset into account in the analysis. This explanation now appears at the beginning of section 3.

For example, It seems clear that the assessment algorithm (starting line 232) can be used to rank the models in terms of their closeness to observations and dynamics in particular regions.

One downside however, is that it does not give the modeller an intuition into how far the model is from "good" behaviour in absolute terms. We simply get an averaged "skill score" from 28% to 79%, but without an idea of how this might relate to more traditional measures of skill. So how close is the best model and how far is the worst model from reality? We have only a score (useful as that is) to guide us.

In this paper we give a global index that is the mean of 7 indices associated with the seven Region-clusters. This mean index shows the ability of a given model to represent the global area. But for each Model and each Region-Cluster, we give the ratio that represents how that model represents that Region-cluster. These indices are visible on the colorbar of figure 3.So each modelling group can evaluate how its numerical schemes represent the dynamic of the observations.

We also give a visual interpretation of the fitting of the different CMIP5 models with respect to observations in Figure 4. The best models are the closest to the observations with respect to the khi2 distance.

In general, we agree with the reviewer that there is a huge amount of information to take out from this method in general and from Fig. 3 and 4 in particular. Each modelling group could use this information to better understand the reasons for their model's difference to observation. And specialists of the Senegal-Mauritania region can use it to better understand the reasons for the

weak representation of this region in climate models. In this paper we propose an illustration of the method application. After publication, the method will be free of use for more various applications. We thus added a sentence in the text as well as in the conclusion to stress that point. We are grateful to the reviewer for having stressed that point.

The paper makes the claim that it offers an objective method for the assessment of the behaviour of models with regards historical observations. I struggle to accept this, given the number of subjective choices made with regards to the way the analysis is conducted. Subjective judgements will always need to be made in the analysis of climate model output - this is inevitable, and perfectly reasonable as long as labelled as such. The paper only examines a subset of model fields for example, and a subjective choice as to which of those fields to select has been made.

The reviewer is right in several aspects:

- The study focuses on the ability of CMIP5 models to reproduce the ocean seasonal variability in the Senegalo-Mauritanian upwelling region only. The models which represent this region at best do not necessarily represent other regions at best. Our study is not devoted to the comparison of the CMIP5 models in general but to their ability to reproduce the Senegal-Mauritanian upwelling area only. We now mention that point explicitly in the conclusion. Furthermore, a full representation of the geophysical phenomenon should involve more variables, as explained in Sylla et al (2019) and this first study is more a test-case than a full analysis.

- Concerning the use of statistics, we are aware that statistics are only a support to understand or interpret what is hidden in the dataset, mainly if the number of data and observations is large. This is the present case because we want to compare 47 different models with respect to a set of observations, with a focus on the dynamical behavior of each dataset (multidimensional analysis in a 12-dimensional space, the monthly SST anomalies). We built a method to solve that problem. Other methods could possibly lead to different results depending of what we looked for: The number of possible statistical studies is huge. In that sense, the choice of the method contains some subjectivity.

Nevertheless, the method we propose is not subjective ; it allows to rank the models according to the reduction of information we made (the seven dynamical region-clusters, after the SOM). This is in some way a classical problem in geophysics, where we need to classify, organize the information. Here it is done using relatively novel statistical tools (SOM+HAC). Finally, the MCA is a qualitative (but rational) method to summarize and visualize on a graphic the "similarities" of the models, the observations and the region-clusters.

For these reasons, we consider that the title is not misleading: our method is a step *towards* an objective assessment. Yet, considering the reviewer's remark, we have modified the sentence claiming for an "objective method" in the abstract and this term is now better justified in the conclusion section.

A core problem that needs to be addressed in the paper can be illustrated by considering the section starting on line 285:

"As indicated in the introduction, the main objective of the methodology is to select an ensemble of models that represents at best the upwelling behavior with respect to the observations and to use this ensemble to predict the impact of climate change in the Senegalo-Mauritanian upwelling with some confidence. The problem is now to determine a subset of models that can adequately represent the observations, as the number of models is small enough we choose to cluster them by HAC according to their projections onto the seven axes provided by the MCA, and select the optimal jump in the hierarchical tree (Jain and Dubes, 1998)."

I cannot see a description of what it means for a subset of models to "adequately represent the observations".

We agree with the reviewer that the phrase "adequately represent the observations" is misleading.
Through MAC+HAC, we group the models into Model-clusters, using the khi2 distance, according to their proximity to the observations and their internal similarity. Model group 4 appears as the one closest to observations with respect to that distance. In Figure 4, we see the projection of the individual models on the first two axes of the MCA. The fact that only two axes are shown here  can introduce some bias in the visualization and this figure must be considered with some caution. We associated a multi-model  with the Model-group 4 (close to the observations),whose outputs are the mean of the outputs of the models constituting the Model-group 4. We agree with the fact that we cannot prove that this is the best. An exhaustive research in order to find the best subset is nevertheless prohibitive due to the enormous number of possible combinations.  The phrase "a subset of models that can adequately represent the observations" was changed into "a subset of models which has a better skill than Model-All".

 I also cannot see an adequate description for what the "optimal jump in the hierarchical tree" of Jain and Dubes (1998) is, or what it might mean for the ensemble members. The clustering of the models in figure 4 looks reasonable by eye, but there are a large number of other ways that the models could be clustered that might be equally as reasonable. The authors claim that their algorithm selects a number of ensemble members that best represent an ensemble mean. I don't believe that they provide sufficient justification for why the ensemble mean should be selected for, or that the ensemble members their algorithm selects members in a way that is superior to a subjective selection.

We recall that the HAC (hierarchical ascending clustering) is a bottom-up algorithm for dataset clustering. The key operation in hierarchical bottom-up clustering is to repeatedly combine the two nearest (according to a certain distance) clusters into a larger  cluster. The HAC starts from individuals and combines them according to their similarity (with respect  to  the  chosen distance) to obtain new clusters.The process is repeated up to get one cluster only (the full dataset). This algorithm is visualised by a tree-like diagram, the so-called connection tree : the

connections between the clusters are represented  by the branches of the connection tree (see figure below) according to their proximities. Due to the bottom-up algorithm,  the construction of clusters is therefore objective with respect to the chosen distance.  The objects are finally categorized  into a hierarchy similar to a tree-like diagram which is called a dendrogram (see figure). A major problem then arises: When do I stop combining clusters and consider that I have optimal clusters?

The problem is semi-qualitative. It depends on the dataset under study. Most of the time, it is a compromise between a sufficient number of clusters to explain the complexity of the dataset and a relatively small number of clusters in such a way that every cluster can be handled and explained.

In the present study, we decided to deepen the statistical aspect of the problem and to choose an "optimal" model according to the data provided by the MCA algorithm (the data are the rows of the matrix   $\mathbf{R} = [\mathrm{R}m i]$ representing the 7 component vector-skill of the models). The HAC clusters the models according to their similarities (based on the  Khi2 distance). The HAC used in the MCA analysis yields the following dendrogram (not shown in the paper):

[Figure]

*Figure: HAC dendrogram*

On the horizontal line, we have displayed the 47 CMIP5 models, each model being associated with its 7 component skill-vector.  As the dendrogram represents  a hierarchy of clusters,  the numbers on the y axis give the distance between two clusters;  Clearly there is an optimal jump on this graph: for 4 clusters we obtain well separated Model-groups that are very different.  The horizontal black line materializes this optimal jump on the figure (level 1.5 in the vertical axis).. The purpose of this explanation is to highlight the rationality of the selection. We reckon that there is subjectivity in the choice of the approach of the statistical tool, and also in the use of the geophysical knowledge of the region. But by themselves, these tools rely on rational criteria.

. This is presented as a "model weighting" paper, and while that might be possible with this algorithm, I do not believe that is where the strength of the analysis lies.

It was not our intention to present this study as a "model weighting" one. Although model weighting strategies are indeed presented in the introduction, the text only refers to model selection. We have carefully proofread the text so as to make sure to remove this misleading message
We indeed agree with the fact that we do not determine a weighted ensemble model but an ensemble model that better represents the observations than Model-all, which is the mean of all the CMIP5 models we have considered, and also better than the other Model-groups. Our model selection is based on the distance separating the models to the observations .
This combination is provided by the MCA which deals with the 7 component skill-vector associated with each model (and permits to determine a distance to the observation field also associated with a 7 component skill vector) which is more informative than the average skill which has one component only.
Our paper is a "model selection" one.

The paper would be better re-cast as a model analysis paper, using an interesting and useful algorithm to explore the dynamical deficiencies of the models in the region, and informing climate modellers of those deficiencies. I think if the authors wish it to be a model weighting paper, then more emphasis needs to be given to the meaning and justification of the weighting scheme. Further, the authors should develop placing the weighting scheme in the context of established work on the meaning of multi-model ensembles.

We agree with the reviewer that our methodology provides rich and objective information about climate models performance in a specific region. This is one outcome of our study (mainly section 3) and we have strengthened this message in the text and in the conclusion.
Nevertheless, We do not agree with the suggestion of the reviewer that the paper is a model analysis paper. Section 4 indeed provides a way (through the MCA) to use a 7 component skill vector to obtain an efficient combination of climate models leading to an efficient multi-model. (Efficient means that its skill is better than the one of Model-all).
Another question may arise, which is far beyond the objective of the present paper: Is model weighting the best strategy to obtain the most efficient multi-model? Or should we envisage statistical combinations based on multi-parameter analyzes as those developed in the present study?.
In our view, the major contribution of our paper can be summarized into the following sentences included in section 7 (discussion and conclusion):

"The extraction of information embedded in the vector-skill whose 7 components are the skills associated with the 7 sub-regions and the resulting efficient multi-model combination imply the use of advanced statistical tools such as the MCA. Moreover, the study of the vector skill also permits to separate information provided on large offshore ocean circulation from those

occurring in the upwelling region leading to diagnose the deficiencies of some climate models with respect to the modelling of physical processes. Another contribution of the MCA is the visualization of the 47 models and the observations on the  plane constituted by the first two MCA axes, which represents 70% of the information embedded in the data.  The similarities of the climate models with respect to the observations and the region-clusters are well evidenced. The 'mean' skill associated with each climate model and proposed in this study is easy to use but is far less informative than the vector-skill whose 7 components are the skills associated with the 7 sub-regions. "

We would like to thank again the reviewer for these comments that helped us improve the paper.

---

## Author Response (AR2)

Response to the  Editor, 2nd round of reviews, Mignot et al GMD 2019-194

We reply here point-by-point to all the reviews. A marked-up manuscript including all the changes follow below.

I concur with the reviewers inference that it is still ambiguous as to whether this paper is focused on a model subsetting method with the Senegalo-Mauritanian upwelling system is a case study, or whether it is a validation paper on the Senegalo-Mauritanian upwelling system performance in CMIP5 which happens to use a certain methodology. It is clear from your response (and it was my initial interpretation) that you consider it to be the former. To make this distinction clearer, I would suggest a more substantial reorganisation of the manuscript, beginning with the discussion about model skill/subsetting starting on P2 L10, then once all of this discussion is out of the way, introduce the upwelling system specifically as a case study.

You are right that our aim is clearly the former. We have reorganized the introduction section following the editor's suggestion in order to make this point clearer. It is now explicated also at the beginning of the conclusion section.

Your response to Reviewer 1's comments 'Give a brief summary of the…' and 'More detail is needed on the ERSST_v3b…' are very light. They do not do as the reviewer asks, which may be appropriate (for example if you are trying to move away from too much focus on the case study, and stick to the technique), but you need to justify this. For example, in your response 'Yet, sensitivity of the method to the target field should definitely be addressed in the future' why not here? Please explain why it is not important, or do it.
We have expanded our responses to Reviewers1 first two comments:

*Lines 22-23: Give a brief summary of the main findings on the future behavior of the Senegalo-Mauritanian upwelling in the abstract, rather than simply stating that the future behavior was assessed.*
*Initial answer: This sentence was changed into* "The future reduction of the Senegalo-Mauritanian upwelling proposed in recent studies is then revisited using this multi-model selection."
The initial response was only concerning the abstract. The results of this previous sentence are now explained at the end of the introduction, when the case study is described. This also helps to better set the problematic of the paper. We are grateful o the reviewer and the editor for insisting on that point.

*Lines 106-107: More detail is needed on the ERSST_v3b data set. How is this dataset produced, and why was it chosen as the "observation field" for comparison with the Models?*
*Initial answer: Some details on the ERSST_v3b were added. We also explained that other data sets would have been available, in particular HadISST. A previous study has shown that differences in the Senegal-Mauritania upwelling are relatively weak. Yet, sensitivity of the method to the target field should definitely be addressed in the future, we thank the reviewer for this remark.*
Our point is that in Sylla et al (2019), we have found that the model's biases and the model's differences were much larger than the differences between the two SST data sets that had been tested. Therefore, the exact dataset chosen here is in fact of little

importance. Even if a sensitivity study could be carried in the future, it is not relevant for the present methodological presentation. This point I now clarified I the text.

I am convinced by your response to Rev2's comments 'A core problem that needs to be addressed in the paper can be illustrated by considering the section starting on line…' and 'I also cannot see an adequate description for what the…' but please integrate these responses fully into the manuscript.
We have added large portions of our response to reviewer 2, including the figure. We hope that the paper is clearer now. Thanks for this suggestion.

P3 L3 'Nevertheless, these models constitute a fully independent ensemble'.
It is not correct to consider the CMIP models independent in a generic sense.
Indeed, the editor is right, this sentence is ambiguous. It has been removed.

As was alluded to by one of the reviewers, the language use is on occasion not succinct and precise or contains mistakes. I highlight a small number of examples of this below, but the manuscript needs to go through a comprehensive copy editing process. GMD does offer this:
https://publications.copernicus.org/services/copy_editing_for_english.html but please don't feel obliged to use this service if you co do this yourself or by different means.

Thanks for this offer. A native English colleague has read through the text. We hope that the language is now acceptable.

. P1 L17 'an efficient multi-model' What does efficient mean in this context?
We have expanded this sentence into "a multi-model ensemble combination that efficiently reproduces target features of the observations" to clarify our point

. P1 L20 'combination of 12' implies that the models within those 12 offers something unique or are ordered in some way. This perhaps plays into the reviewer's view that you were trying to weight models rather than subset models. It maybe that you are suggesting they offer something unique, but my interpretation here is that you mean 'subset'.
The editor is right, and we have corrected this flowing his suggestion.

[revised manuscript text omitted]

ACCESS1-3
CESM1-CAM5
CESM1-CAM5-1-FV2
CESM1-WACCM
HadCM3
MIROC-ESM
MIROC-ESM-CHEM
MIROC5
NorESM1-M
NorESM1-ME | bcc-csm1-1
bcc-csm1-1-m
BNU-ESM
CCSM4
CESM1-BGC
CESM1-FASTCHEM
GFDL-CM2p1
GFDL-ESM2G
GFDL-ESM2M
MPI-ESM-LR
MPI-ESM-MR
MPI-ESM-P | FGOALS-g2
GISS-E2-H
GISS-E2-H-CC
GISS-E2-R
GISS-E2-R-CC
inmcm4
IPSL-CM5A-LR
IPSL-CM5A-MR
IPSL-CM5B-LR
MRI-CGCM3
MRI-ESM1 | CanCM4
CanESM2
CMCC-CESM
CMCC-CM
**CMCC-CMS**
**CNRM-CM5**
**CNRM-CM5-2**
CSIRO-Mk3-6-0
**FGOALS-s2**
**GFDL-CM3**
HadGEM2-AO
HadGEM2-CC
HadGEM2-ES |

| ZModel-group 1 | ZModel-group 2 | ZModel-group 3 | ZModel-group 4 |
|---|---|---|---|
| ACCESS1-0
bcc-csm1-1-m
CCSM4
CESM1-BGC
CESM1-CAM5
CESM1-CAM5-1-FV2
CESM1-FASTCHEM
CESM1-WACCM
GISS-E2-H
GISS-E2-H-CC
GISS-E2-R
GISS-E2-R-CC
HadCM3
inmcm4
IPSL-CM5B-LR
MIROC5
MPI-ESM-LR
MPI-ESM-MR
MPI-ESM-P | **CMCC-CMS**
**CNRM-CM5**
**CNRM-CM5-2**
**FGOALS-s2**
**GFDL-CM3** | BNU-ESM
CanCM4
CanESM2
CMCC-CM
FGOALS-g2
IPSL-CM5A-LR
IPSL-CM5A-MR
MRI-CGCM3
NorESM1-M
NorESM1-ME | ACCESS1-3
bcc-csm1-1
CSIRO-Mk3-6-0
HadGEM2-AO
HadGEM2-CC
HadGEM2-ES
MIROC-ESM
MIROC-ESM-CHEM
MRI-ESM1 |
| | | | **ZModel-group 5** |
| | | | CMCC-CESM
GFDL-CM2p1
GFDL-ESM2G
GFDL-ESM2M |

[revised manuscript text omitted]